# Perinatal inflammation influences but does not arrest rapid immune development in preterm babies

S. Kamdar[1], R. Hutchinson[2,3], A. Laing[1], F. Stacey[2], K. Ansbro [3,5], M.R. Millar[3], K. Costeloe[3], W.G. Wade [4], P. Fleming[2,3,6] & D.L. Gibbons [1,6✉]

Infection and infection-related complications are important causes of death and morbidity following preterm birth. Despite this risk, there is limited understanding of the development of the immune system in those born prematurely, and of how this development is influenced by perinatal factors. Here we prospectively and longitudinally follow a cohort of babies born before 32 weeks of gestation. We demonstrate that preterm babies, including those born extremely prematurely (<28 weeks), are capable of rapidly acquiring some adult levels of immune functionality, in which immune maturation occurs independently of the developing heterogeneous microbiome. By contrast, we observe a reduced percentage of CXCL8-producing T cells, but comparable levels of TNF-producing T cells, from babies exposed to in utero or postnatal infection, which precedes an unstable post-natal clinical course. These data show that rapid immune development is possible in preterm babies, but distinct identifiable differences in functionality may predict subsequent infection mediated outcomes.

[1] Peter Gorer Department of Immunobiology, King's College London, London SE1 9RT, UK. [2] Department of Neonatology, Homerton University Hospital, London, UK. [3] Barts and the London School of Medicine and Dentistry, Queen Mary University of London, London, UK. [4] Faculty of Dentistry, Oral and Craniofacial Sciences, Centre for Host-Microbiome Interactions, King's College London, London, UK. [5] Present address: University of Sheffield, Sheffield, UK. [6] These authors contributed equally: P. Fleming, D. L. Gibbons. ✉email: deena.gibbons@kcl.ac.uk

Longitudinal data consistently show overall mortality following preterm birth to be falling; however, among babies who die, the proportion attributed to infection and infection-related complication such as necrotising enterocolitis (NEC) is fixed or increasing[1,2]. It is estimated that up to 30% of preterm babies born before 32 weeks gestation develop at least one clinical or microbiologically confirmed episode of sepsis[3], but the course of individual babies through the perinatal period is unpredictable, some progressing smoothly while others succumb to multiple pathologies, including late-onset sepsis and NEC.

There is a clinical need both to identify around the time of birth those preterm infants most likely to develop infective complications and to develop therapies to prevent and ameliorate infection in this vulnerable group, ideally through promoting the baby's own immune capabilities. Understanding the development of the immune system in the healthy preterm infant, how it differs in those with complicated courses and how it is modified by peri- and postnatal factors is fundamental to these aims. Furthermore, not only are these issues acutely relevant to the infant, but increasing evidence suggests that the nature of the infant immune response in early life, where rapid immune development has been observed[4,5], may profoundly influence later susceptibility to immune mediated diseases[6,7]. Thus, infants born before 32 weeks gestational age (GA) show a greater susceptibility to asthma in later life and yet a decreased risk of developing food allergy[8,9].

Despite these drivers, understanding of how the preterm immune system develops is incomplete, and the impacts on the developing immune system of exposure to infection in utero, as well as the postnatal clinical course, have not been comprehensively evaluated.

It is increasingly acknowledged that the preterm immune system is not just an immature version of the adult, but is both qualitatively and quantitatively distinct with overt functional potentials[10]. To address how the immune system develops in those born prematurely, we prospectively and longitudinally follow a cohort of preterm babies born before 32 weeks gestation and compare their immune function for up to 3 months post birth. We demonstrate that preterm babies (even those born at the earliest gestation) rapidly acquire some adult levels of immune function; have immune functions that progress towards but do not reach adult levels; or have immune functions, conspicuously C-X-C motif chemokine ligand 8 (CXCL8)-producing T cells, that remain significantly distinct from adults. Babies born at an earlier GA show immune profiles that change more dramatically when compared to those born at later gestations, thus reducing any negative impact of extreme prematurity. Immune development occurs independently of specific changes within the developing microbiome, suggesting that other factors may have a greater bearing on immune development in the first few months of life. A defining feature of those infants who develop infection is a lower level of CXCL8-producing T cells initially, thus identifying a potential biomarker for identifying infants at greater risk of sepsis. This, in turn, may help reduce the burden posed by infection and lead to improved outcomes for babies born prematurely.

## Results

**Patient demographic and clinical characteristics**. Thirty-nine babies with a median gestation of 28.7 weeks (range 23.6–31.7) and a mean (SD) birthweight of 1060 g (383) enrolled between January 2016 and December 2017 were assessed. Participant demographic and clinical data are provided in Table 1. Ten babies were classified as stable (receiving a single course of antibiotics in the immediate postnatal period) and 16 were unstable, having

received additional courses of antibiotics for suspected late-onset sepsis or other infection concerns including NEC. Thirteen further infants were born to mothers with histological or clinically confirmed chorioamnionitis, all but one of these also fulfilled the unstable criteria. Of note, babies in the stable group were generally born at later gestation, had higher birth weights and all were delivered by caesarean section.

**Postnatal immune development in all babies**. One hundred and eighty-six different peripheral blood immune populations were analysed, with and without in vitro stimulation, using 11-colour flow cytometry panels. Samples were taken longitudinally (mean of eight samples per baby) from the infant cohort until discharge from the neonatal intensive care unit (NICU) or until 36 weeks post-menstrual age. The sampling regime, analysis pipeline and clinical groupings are depicted in Fig. 1. Initially, we identified immune parameters that were most distinct between adults and preterm infants within the first few days of birth (Supplementary Fig. 1). Unsurprisingly, CXCL8 (a cytokine known to be pivotal in this demographic[10]) was one of the most divergent parameters, significantly elevated in infants compared to adults.

**Changes in immune parameters over time**. We then studied any alterations in these immune parameters over time. To address which parameters changed most significantly with postnatal age, we compared the regression coefficients of each parameter in each baby and plotted the mean value in relation to how different they were from adult at initial sampling (Fig. 2a). Over half of all parameters studied showed significant changes in this time period. Many parameters that were reduced around birth (anything below the half way line on y-axis) increased (and hence appeared in lower right-hand quadrant), and conversely those that were elevated in the first sample taken from the neonate (anything above the half way line on y-axis) decreased towards adult levels —located in the upper left-hand quadrant. Conspicuously, although CXCL8-producing natural killer (NK) and γδ cells that were initially elevated subsequently decreased with time, this was not true for CXCL8-producing CD4 and CD8 T cells of various phenotypes. These populations started elevated compared to adults and remained so, highlighting their potential functional importance in the neonatal immune response.

Parameters that increased over time included B cells, which increased in both percentage (Fig. 2b) and absolute number (Supplementary Fig. 2a); NK cells that increased in both percentage (Fig. 2c) and absolute number (Supplementary Fig. 2b); γδ T cells (Supplementary Fig. 2c); HLADR expression on monocytes (Fig. 2d and Supplementary Fig. 2d); and NKG2D expression on NK cells (Fig. 2e). Consistent with the observation that the CD4:CD8 ratio is much greater in (term) neonates than adults[11], the percent representation of CD8$^+$ T cells increased over time coincident with an increase in their absolute number (Supplementary Fig. 2e, f). CD161 expression on CD8 T cells decreased in the 3-month study period (Fig. 2f).

With respect to changes in functionality that occurred postnatally (determined by assessing cytokine production following in vitro stimulation with phorbol myristate acetate and ionomycin), we observed significantly increased percentages of interferon-γ (IFN-γ)-producing NK cells (Fig. 2g) and γδ T cells (Fig. 2h and Supplementary Fig. 3a). However, levels of IFN-γ-producing CD4 T cells in these babies remained extremely low even at 3 months of age and there was no reproducible increase in the capacity of CD4 T cells to produce IFN-γ over time (Fig. 2i and Supplementary Fig. 3b). CD4 T cells continued to produce CXCL8, however, and the levels did not diminish towards that seen in adult in the time frame studied (Fig. 2j). We observed that

**Table 1 Participant demographic and clinical data.**

| | Total babies (n = 39) | Stable babies (n = 10) | Unstable babies (n = 16) | Chorioamnionitis (BCM, n = 13) |
|---|---|---|---|---|
| Gestation, median | 28.7 (23.6–31.7) | 30.5 (29.1–31.7) | 28 (23.6–31.7) | 25.7 (24–30.1) |
| Birthweight, mean | 1060 g (383) | 1366 g (242) | 1062 g (422) | 821 g (241) |
| Sex | | | | |
| Male | 23 (59%) | 6 (60%) | 11 (69%) | 6 (46%) |
| Female | 16 (41%) | 4 (40%) | 5 (31%) | 7 (54%) |
| Multiple births | | | | |
| Singleton | 29 (73%) | 5 (50%) | 11 (69%) | 13 (100%) |
| Multiple | 10 (27%) | 5 (50%) | 5 (31%) | |
| Apgar 5 min, median | 9 (2–10) | 10 (6–10) | 9 (2–10) | 9 (2–10) |
| Maternal ethnicity | | | | |
| White | 9 (23%) | 1 (10%) | 4 (25%) | 4 (31%) |
| Black | 20 (51%) | 4 (40%) | 10 (63%) | 6 (46%) |
| Asian | 7 (18%) | 4 (40%) | 1 (6%) | 2 (15%) |
| Other | 3 (8%) | 1 (10%) | 1 (6%) | 1 (8%) |
| Antenatal steroids | | | | |
| Any | 36 (92%) | 10 (100%) | 13 (81%) | 13 (100%) |
| >24 h before delivery | 18 (46%) | 6 (60%) | 7 (44%) | 5 (38%) |
| None | 3 (8%) | 0 (0%) | 3 (19%) | 0 (0%) |
| Delivery by caesarean | | | | |
| Yes | 24 (62%) | 10 (100%) | 12 (75%) | 2 (15%) |
| No | 15 (38%) | 0 (0%) | 4 (25%) | 11 (85%) |
| Chorioamnionitis | 13 (33%) | 0 (0%) | 0 (0%) | 13 (100%) |
| Histologically confirmed | 12 (31%) | | | 12 (92%) |
| Antenatal antibiotic administration | 11 (28%) | 4 (40%) | 2 (12%) | 5 (38%) |
| Microbiologically confirmed sepsis (any episode) | 14/39 (36%) | 0 (0%) | 7 (44%) | 7 (54%) |
| Days antibiotics, median, range | 15 (3–51) | 4 (3–7) | 19 (5–51) | 19 (5–35) |
| Suspected GI pathology (including NEC)[a] | 7/39 (18%) | 0 (0%) | 5 (31%) | 2 (15%) |
| NEC Bell's stage II or > | 3/39 (7%) | 0 (0%) | 3 (19%) | 0 (0%) |
| Days to full enteral feeds[b] | 14 (7–42) | 12 (7–37) | 16.5 (8–33) | 15 (7–42) |
| Received any maternal breast milk | 37/39 (95%) | 10 (100%) | 14 (87.5%) | 13 (100%) |
| Abnormalities on CrUSS | | | | |
| IVH[c] | 6 (15%) | 1 (10%) | 2 (12.5%) | 3 (23%) |
| HPI | 2 (5%) | 0 | 1 (6%) | 1 (8%) |
| PVL | 3 (8%) | 0 | 1 (6%) | 2 (15%) |
| Chronic lung disease[d] | 14 (36%) | 0 (0%) | 8 (50%) | 6 (46%) |
| ROP[e] | 16 (10%) | 1 (10%) | 8 (50%) | 7 (54%) |

Data are presented as means (standard deviation); medians (ranges); n = number and percentages.
*CRUSS* cranial ultrasound, *IVH* intraventricular haemorrhage, *HPI* haemorrhagic parenchymal infarction, *PVL* periventricular leukomalacia, *CLD* chronic lung disease, *ROP* retinopathy of prematurity.
[a]Suspected GI pathology is defined as 'any abdominal concerns necessitating nil by mouth for more than 5 days'.
[b]Full enteral feeds defined as 'the first day on which 100% of fluid volume was administered enterally'.
[c]IVH is defined as 'bleeding contained within and not extending beyond the ventricular system'.
[d]CLD is defined as 'the need for supplemental oxygen at 36 weeks post-menstrual age'.
[e]ROP is defined as 'any stage ROP recorded on formal eye examination'.

the time to blood processing did have an adverse effect on cytokine production. Nevertheless, when we assessed any relationship between processing time and the sample age (considering we are assessing changes in parameters over time) there was none, further highlighting our data that suggest that the development of immune parameters is related to postnatal age

**Effect of extreme prematurity**. To identify whether the consensus pattern of immune development that we observed in our infant cohort also occurred in the event of extreme prematurity, we performed principal component analysis (PCA) analyses using all the flow cytometry parameters to generate immune profiles for each baby at each time point. By comparing the immune profile generated in the first week of life compared to that generated at around 5 weeks of age, we observed that the babies' immune profiles progressed in a similar direction as they aged (Fig. 3a). Importantly, this trajectory was generally followed by all the babies, independent of the infant's GA at birth. In fact, when we examined distance travelled along this trajectory, babies born at the earliest GA tended to exhibit a greater degree of movement over a similar time period to those born at later GAs (Fig. 3b). *t*-Distributed stochastic neighbour embedding analyses suggested that even though all babies progressed down this shared developmental pathway, samples taken from the same baby (represented by individual colours) appeared to retain unique features sufficient to cluster samples most closely according to their individual profile. This differentiation of babies one from another

was sustained throughout their time in the NICU (Fig. 3c). Due to ethical considerations around obtaining detailed longitudinal samples from term infants, we were unable to assess inter-individual immune profile heterogeneity in a term infant cohort. We were, however, able to demonstrate a similar trajectory in immune parameters in samples obtained from different term infants (n = 6) taken at different postnatal ages (up to 3 months) despite their inherent heterogeneity. Functionality (as assessed by cytokine production) increased in γδ T cells in both term and preterm cohorts, whereas both cohorts failed to show a reproducible increase in the ability of CD4 T cells to produce IFN-γ. This suggests that postnatal adaptation may be similar, irrelevant of GA at birth and that many immune parameters develop after birth. Whereas the general direction of immune development was mirrored in the preterm and term infants, there were some exceptions, notably CD161 expression on CD8 T cells. The percentage of CXCL8-producing CD4 T cells also appeared to decrease faster in those born at term compared to those born prematurely (Supplementary Fig. 4a).

**Associations between immune parameters**. To identify any immune cell populations that may associate with each other (either positively or negatively) in our infant cohort, we performed Spearman's correlation coefficients ($R \geq 0.3$ or $R \leq -0.3$) across our whole data set. We identified significant associations ($p < 0.008$) between several immune parameters. These associations were grouped into nodes based on cell type (see Fig. 3d).

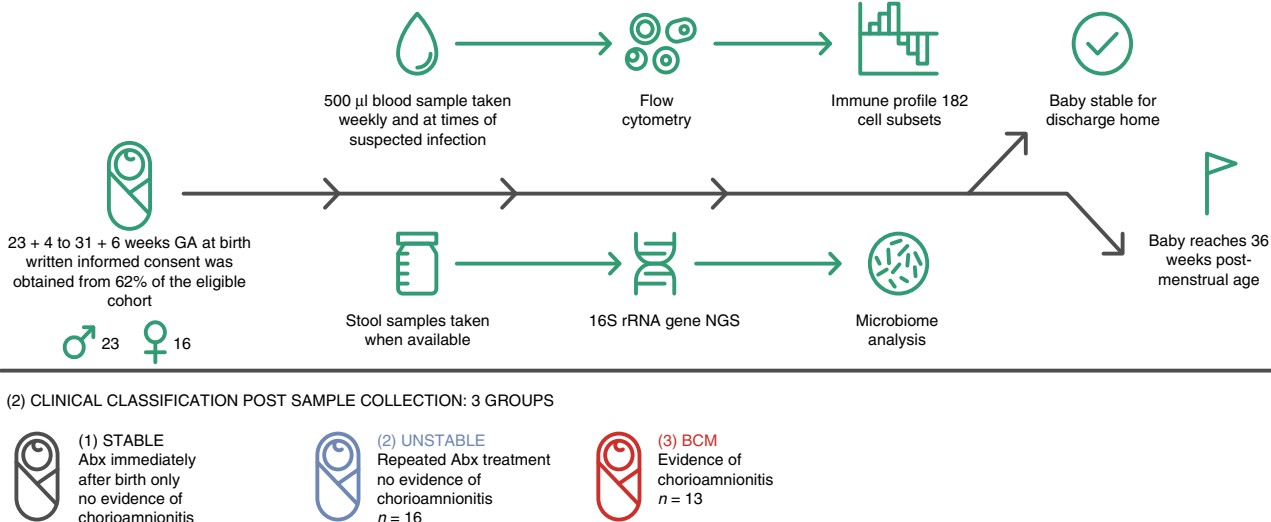

**Fig. 1 Schematic representing study design.** Written informed consent was obtained from parents before 72 h of age and of the parents approached, 62% agreed to enter the study as described in the figure.

Some populations (e.g. intermediate monocytes) showed no correlation with others and were subsequently lost from this network analysis. Due to the linked nature of flow cytometry, we only used one parameter (of a correlated pair) in the analysis to avoid correlations due to experimental technique. Examples of unanticipated positive correlations included the expression of CD161 on γδ T cells correlating with their production of IFN-γ ($R = 0.5$) and between IFN-γ-producing (CXCL8−) γδ and CD8 T cells ($R = 0.41$) (Fig. 3e). There was also a good correlation between CXCL8-producing CD4 and CD8 T cells ($R = 0.62$) (Fig. 3e). Similar correlations were also observed in term babies ($n = 9$) under 6 months of age (Supplementary Fig. 4b).

**Potential sources controlling immune profile variation.** There were extensive differences identified between the neonatal preterm infant and the equivalent adult immune profile. However, while some immune parameters exhibited rapid maturation over the time frame of our study, even in those infants born extremely preterm, others showed little change. Considering the links previously suggested between immune cell development, immune homeostasis, and the microbiome[12], we considered the developing intestinal microbiome as a potential driver of these changes in immune profile. Similarly, many immune parameters are sexually dimorphic, at least in mice[13]. Hence, we also considered the influence of sex on these developing immune profiles.

Stool microbiome analyses were performed on 10 stable infants (100% of eligible cohort); 13 unstable infants (81% of eligible cohort); and 11 infants born to mothers with chorioamnionitis (85% of eligible cohort). Initially, we analysed the development of the microbiome by assessing the range of different taxa that appeared in faeces over time (longitudinal development of diversity). In the stable group, microbiome diversity increased at a rate of 0.56 U per week (95% confidence interval: 0.3–0.9). Conversely, there was significantly restricted growth in diversity in both the unstable (median 0.035 vs. 0.56 U per week, $p < 0.0001$) and the chorioamnionitis group (0.053 vs. 0.56, $p < 0.0001$), with no significant difference in the rate of diversity progression between these cohorts (Fig. 4a).

When assessing the emergence of taxa in our cohort, the family *Enterobacteriaceae* was the dominant family across all the groups.

Figure 4b shows representative examples of taxa progression in individual babies from the three clinical classifications. When the longitudinal progression of *Enterobacteriaceae* was compared between the groups, *Enterobacteriaceae* relative abundance fell much quicker in the stable cohort (median 5.6% per week) than either the unstable (−5.6 vs. 0.16, $p = 0.04$) or the chorioamnionitis groups (−5.6 vs. 0.38, $p = 0.06$) (Fig. 4c).

As *Enterobacteriaceae* relative abundance fell in the stable cohort, a shift towards higher proportions of other families was observed. The next most prevalent were *Veillonellaceae* (which gradually increased over time) and *Bifidobacteriaceae* (which increased from ~day 30) having a maximum relative abundance of >1% in 9/10 and 10/10 infants, respectively (Supplementary Fig. 5). Conversely, *Staphylococcaceae* were highly prevalent (>1% maximum abundance in 8/10 participants) in early life, becoming almost absent beyond day 20. Other notable families (*Clostridiaceae* 8/10; *Enterococcaceae* 9/10) changed minimally over time, and generally maintained low median abundances. In the unstable groups, these consistent patterns of microbial community progression were not observed (Supplementary Fig. 5), although a strikingly decreased prevalence of *Bifidobacteriaceae* was noted (46% of unstable infants were never colonised with *Bifidobacteriaceae*, that is, never had a relative abundance in any sample >1%).

We then analysed the microbiome structure in conjunction with the immune parameters. Each microbiome profile was compared to each immune profile in individual babies and the data combined to identify any significant correlations across the 34 infants on which both sets of data were available. We could identify very few correlations between the two data sets (Supplementary Fig. 6). The only weak association identified was between *Staphylococcaceae* and myeloid dendritic cells (DCs).

To establish additional factors driving the different immune profiles in our preterm infant cohort, we analysed for differences driven by sex. Perhaps surprisingly, the majority of immune parameters showed no significant sexual dimorphism, but there were three notable exceptions. Both CD31 and CD38 expression on CD4 T cells were significantly elevated in female infants compared to their male counterparts (Supplementary Fig. 7a, b). In contrast, CD161 expression on CD8 T cells was higher in males and then subsequently decreased

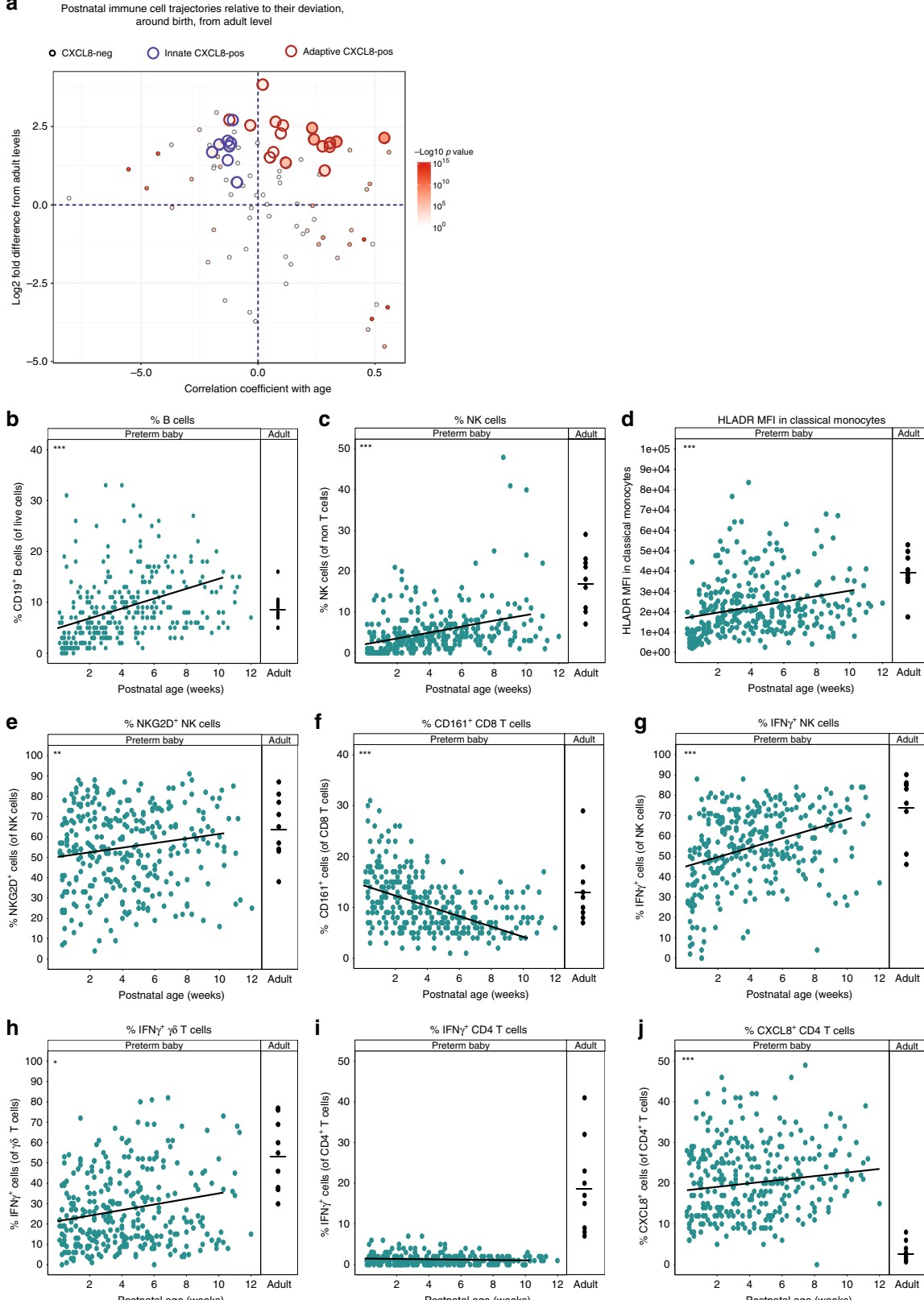

more dramatically in expression over time than in females (Supplementary Fig. 7c).

**In utero and postnatal exposure to inflammation.** Having shown limited associations between immune parameters and sex/

microbiome, we considered the effects of both pre- and postnatal exposure to infection on the developing immune system. Analyses of the whole immune profiles by PCA for the different clinical groups described in Table 1 identified distinct immune signatures in babies who had experienced different pre- and postnatal exposures to the stable infants (Fig. 5a, b). Babies who

**Fig. 2 Phenotypic and functional maturation of distinct immune parameters.** Longitudinal PBMC samples from 39 preterm babies were phenotyped for 186 different immune populations by flow cytometry following surface and intracellular staining. For cytokine detection, samples were activated in vitro with PI (4 h, in the presence of BFA) prior to staining. **a** Postnatal Immune cell trajectories relative to their deviation from the first infant sample taken, compared with adult levels. Position along $y$-axis indicates deviation from adult (the log 2 fold difference) in the first sample; position along $x$-axis indicates whether the population has increased or decreased significantly in the infant over time. Thus, immune populations in the top left quadrant start higher than adult levels and then decrease over time towards the adult level. Populations in the bottom right start below adult levels and increase over time towards that of the adult. Colour indicates—log 10 $p$ value (Wilcoxon's matched-pairs signed-rank test) for correlation with age with large circles indicating CXCL8-producing cell types. Among the CXCL8-producing populations, the red outline indicates NK and γδ T cells, whereas the blue outline CD4 or CD8 αβ T cells. **b–j** Changes in individual immune parameters over time depicted by scatter plots showing frequencies/MFI (as indicated) in preterm babies (left panel; cyan circles) as a function of postnatal age compared to adults (right panel; black circles). **b** CD19[+] B cells, **c** HLADR MFI in classical monocytes, **d** NK cells, **e** NKG2D[+] NK cells, **f** CD161[+] CD8 T cells, **g** IFN-γ[+] NK cells, **h** IFN-γ[+] γδ T cells, **i** IFN-γ[+] CD4 T cells, and **j** CXCL8[+] CD4 T cells. In the figure, cyan circles represent a pool of longitudinal samples from 39 preterm babies where each circle represents an individual sample and on average there are eight longitudinal samples per baby. Black circles are a pool of samples from nine adults. ***$p < 0.001$, **$p < 0.01$, and *$p < 0.05$ as determined by linear mixed-effect modelling using the lmer package in R. Source data are provided as a Source Data file.

had an unstable course or those born in the context of chorioamnionitis showed elevated levels of the activation marker CD69 on all T cell populations (CD4, CD8 and γδ) as well as on NK cells (Fig. 5c–f) when compared to babies who had a stable postnatal course. CD35 (also known to increase upon cell activation) was also elevated on both CD4 and CD8 T cells, significantly so in those infants exposed to chorioamnionitis (Supplementary Fig. 8a). There was no difference in CD25 expression, however, on either CD4 or CD8 T cells between the different clinical groups.

Stable infants exhibited low levels of proliferation initially (<10% of CD4 T cells were Ki67 positive in the first blood sample). However, there was a consistent proliferative burst (to nearly a quarter of CD4 T cells) around 2 weeks after birth, which rapidly returned to post birth levels, seen in nearly all infants (Fig. 5g). Elevated CD4 T cell proliferation was also observed immediately post birth in infants born to mothers with chorioamnionitis (Fig. 5g). Similarly, elevated levels of intermediate monocytes, particularly in the immediate postnatal period, were seen in babies born to mothers with chorioamnionitis ($p = 0.0004$, Fig. 5h), both suggestive of in utero activation. However, levels of HLADR expression on these perinatal monocytes were significantly lower than that seen in stable infants (Supplementary Fig. 8b). FOXP3 expression was significantly reduced on T-regs isolated from infants born to mothers with chorioamnionitis (Fig. 5i). CD8 expression was maintained on γδ T cells (both Vδ1 and Vδ2) in unstable infants compared to those with a stable clinical course (where levels decreased rapidly post birth, Fig. 5j, k).

To identify any changes in functionality between the different clinical groups, cytokine production following in vitro stimulation with phorbol myristate acetate and ionomycin was assessed. The ability of CD4 T cells to produce tumour necrosis factor (TNF) and interleukin-2 (IL-2) (Fig. 6a, b) upon in vitro stimulation was indistinguishable between the three clinical groups, yet, in contrast, the ability to produce CXCL8 was severely and significantly reduced in both the chorioamnionitis and unstable groups when compared to stable infants (Fig. 6c). It has been previously suggested that the capacity of T cells to produce TNF and CXCL8 was reciprocally regulated based on GA[14]. However, we did not see significant changes in the capacity of CD4 T cells to make TNF or CXCL8 based on their GA at birth in this cohort (Supplementary Fig. 9a, b). Similarly, babies born at the same GA exhibited very different percentages of T cells with the capability to produce either CXCL8 or TNF (Supplementary Fig. 9c, d). In every case, despite an equivalent GA at birth, those babies exhibiting the lowest levels of CXCL8-producing T cells (but not TNF-producing T cells) were those infants with an unstable clinical course (Supplementary Fig. 9c, d). Furthermore, when we

separated bacteraemias into those from coagulase-negative staphylococci (CoNS) vs. other bacteria (as CoNS infections generally have lower morbidity and mortality rates), those infants demonstrating the lowest initial levels of CXCL8-producing T cells were those that subsequently developed the more severe (non-CoNS) infections. This was despite an equivalent capacity to produce TNF or IL-2 (Fig. 6d–f). When compared to stable infants, significantly lower CXCL8-producing CD4 T cells were seen in infants with clinically suspected sepsis[3] ($n = 16$, $p = 0.007$) and in those infants with microbiologically confirmed sepsis caused by either coagulase CoNS ($n = 9$, $p = 0.039$) or other bacteria ($n = 4$, $p = 0.0007$) such as Group B streptococcus, *Escherichia* coli or *Streptococcus anginosus*.

## Discussion

The consequences of preterm birth on immune development and function are not well established[15]. In this study, we demonstrate that preterm babies, even those born at 23 and 24 weeks gestation, were capable of rapidly acquiring several components of immune function. The immune profiles for babies born at earlier gestations changed more dramatically over a similar time period to those born at later gestations, which suggests that extremely preterm babies are capable of rapid progression to 'catch up' immune function. This corroborates data suggesting that the immune profiles of preterm and term babies converge in a similar time frame[4] and that rapid immune development in term babies follows a set trajectory[5]. Despite inherent heterogeneity in immune parameters between individuals, longitudinal changes in immune function were observed both in our preterm cohort and in different infants born at term and sampled at different ages during the postnatal period. This suggests similar postnatal adaptation in both cohorts. However, despite this immune development, some of the preterm babies developed infection. Strikingly, this was associated with a reduced T cell capacity to produce CXCL8, highlighting the CXCL8 pathway as a potential therapeutic target for preventing infection in preterm babies. The interpretation of our data is however constrained by the lack of an equivalent longitudinal data set from a cohort of healthy full-term infants to which to compare our findings. In our experience, no research ethics committee would sanction such extensive sampling in healthy term babies to the same degree of sampling that we have conducted in patients recruited to our study. To address this, we, like others, have analysed samples obtained opportunistically from hospitalised infants, in our case awaiting cardiac surgery; the normality of which can be challenged.

Immune development in these preterm infants occurred apparently independently of changes within the developing intestinal microbiome. Prolonged broad-spectrum antibiotic

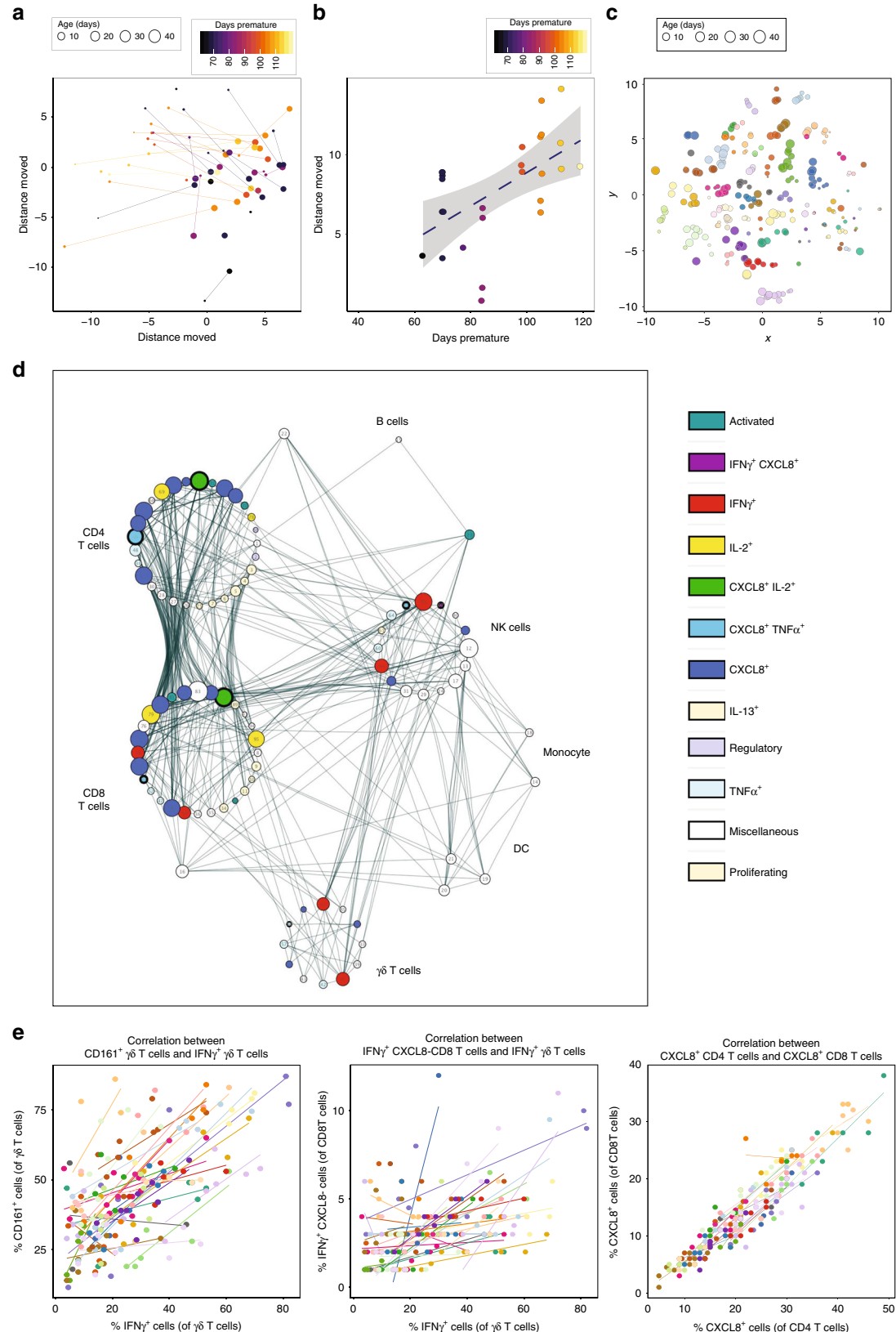

treatment of the unstable infants likely transformed the developing microbiome and yet those infants were still capable of immune maturation along the established trajectory. Given the generally naive nature of the neonatal immune system, it is conceivable that exposure to *any* environmental stimulus is sufficient to drive the observed maturation of the immune system during this period. Indeed, the sudden increase in proliferation of CD4 T cells and other T cells, which we observed at around 2 weeks of age, may be related to microbial colonisation. Rapid intestinal colonisation with a range of organisms would most likely far eclipse any differences driven by specific individual colonisers. Alternatively, microbial–immune relationships may

**Fig. 3 Development of whole immune profiles and correlations.** Longitudinal PBMC samples were phenotyped by flow cytometry as described. **a, b** Analyses of the immune parameters from 39 preterm babies shows that the immune profile of extremely preterm babies (lighter colour) travels a longer distance, but on the same trajectory compared to their less preterm counterparts (darker colour) as depicted by: **a** PCA of the immune profile of samples taken in the first week (small circle, mean = 3.8 days) compared to that of the sample taken 5 weeks later (large circle, mean = 37 days) for each baby. The colour gradient represents how many days preterm the baby was at birth (counting down from 40 weeks). Each circle represents a sample from an individual baby and the size represents postnatal age of the baby at sampling. Lines link individual babies. **b** Scatter plot quantifying the Euclidian distance moved in PC1 and PC2 for each baby. The colour gradient represents how many days preterm the baby was at birth. Each symbol represents an individual baby. **c** t-Distributed stochastic neighbour embedding (tSNE) analysis of the immune parameters shows that babies have their own individual profile. Each circle represents a sample from an individual baby (the size of the circle depicts the postnatal age of the baby when the sample was taken) and each colour represents longitudinal samples from the same baby. **d** Network represents statistically significant ($p < 0.008$) Spearman's correlation coefficients ($R \geq 0.3$ or $R \leq -0.3$) between immune parameters. Each node represents an immune parameter. Nodes are grouped by lineage and coloured by function; node size relates to the number of relationships. If any parameters were inherently related to other parameters due to the nature of flow analysis, only one parameter (of a correlated pair) was used in the correlation plot. **e** Plots depicting correlation between frequency of (left panel) IFN-$\gamma^+$ $\gamma\delta$ T cells and CD161-expressing $\gamma\delta$ T cells; (middle panel) IFN-$\gamma^+$ CXCL8$^-$ CD8 T cells and IFN-$\gamma^+$ $\gamma\delta$ T cells; and (right panel) CXCL8-producing CD4 and CD8 T cells. The data shown are a pool of longitudinal samples from 39 preterm babies, where each circle represents an individual sample and on average there are eight longitudinal samples per baby. Individual babies are in different colours. Source data are provided as a Source Data file.

exist below family level. Similarly, although the stable infants showed some consistency in stool microbial profiles with decreased *Enterobacteriacae* over time, these were replaced by a variety of different obligate anaerobes.

While the mode of birth has been associated with the development of specific microbiomes[16], we were unable to dissociate mode of birth from other exposures as all stable (but not all unstable) infants in our cohort were born by caesarean section. The observed association of high *Staphylococcaceae* abundance in early life in the stable group (uniformly born by operative delivery) is consistent with the previously noted association of staphylococci colonisation after caesarean section (as opposed to natural birth)[17,18]. Indeed, the only (weak) association of the microbiome with the immune parameters was a negative association between *Staphylococcaceae* abundance and myeloid DCs, a proportion of which could be driven by age. Similarly, while breast-feeding (either wholly or partially) is also known to significantly influence microbiome development[19], almost all babies received at least some maternal breast milk.

Aspects of both innate and adaptive immunity developed rapidly during the period of study. B cells increased in numbers after birth as did $\gamma\delta$ T cells whose function (as determined by IFN-$\gamma$ production) was enhanced over time, both in those born prematurely or at term. We identified a decrease in CD8 expression on $\gamma\delta$ T cells, but interestingly, only in stable preterm babies. This suggests either clonal expansion of an existing CD8$^+$ population and/or increased CD8 expression upon activation in those infants experiencing episodes of infection. It is intriguing that we see significant changes in the $\gamma\delta$ T cell population in our cohort as we and others have suggested these cells are of particular importance in providing protection to infants in early life[20–22]. In contrast, $\alpha\beta$ CD4 T cells showed scant, if any, development over this time period, in both term and preterm cohorts. There was little change in their ability to produce CXCL8, which, if anything, increased over the time frame in those born prematurely while decreasing slightly in those born at term, presumably as a result of different levels of T cell export from the thymus[23,24] and they were still unable to produce IFN-$\gamma$ upon ex vivo stimulation. This relative inability to produce IFN-$\gamma$ may be related to hypermethylation of the IFN-$\gamma$ promoter previously observed in human newborn infants[25]. Although we did not see any significant increase in IFN-$\gamma$-producing CD8 T cells over postnatal age, there was a significant association between IFN-$\gamma$-producing $\gamma\delta$ T cells and IFN-$\gamma$-producing CD8 T cells, suggesting some maturation in the latter, albeit at a slower rate than that observed for $\gamma\delta$ T cells.

In this study, an individual baby's subsequent clinical course could not have been predicted at the time of birth. Babies with an unstable postnatal clinical course and those born in the context of chorioamnionitis both succumbed to higher rates of microbiologically confirmed sepsis and chronic lung disease when compared to the stable cohort. Our data highlight the influence of both the in utero environment and the postnatal clinical course on the developing immune profiles of preterm infants. Bacteria implicated in chorioamnionitis (such as *E. coli*) have also been shown to produce histone-modifying enzymes to down-regulate protein expression and there is increasing evidence that sepsis can induce epigenetic changes that alter immune responses[26,27]. We cannot exclude the potential confounder of GA in our data set as all of the stable babies were born at a later GA. This notwithstanding, many of the parameters that differed between the clinical groups seemed less likely to be related to GA as opposed to clinical state. Furthermore, if we plotted all our parameters based on postmenstrual age, the graphs were more or less identical. Hence, an activated immune profile straight after birth, as suggested by several parameters (e.g. expression of CD69, CD35 and proliferation) and previously observed[28], was seen in infants born to mothers with chorioamnionitis. Furthermore, significantly elevated intermediate monocytes were also observed in this group. This is despite the fact that these cells are particularly sensitive to delays in blood processing, suggesting that this may be an under-representation of the true levels. However, although they increased in number, their functional potentials may be poor as we identified significantly reduced HLADR expression on these cells, as has also been observed previously[29]. Indeed, monocytes developing in the context of chorioamnionitis have shown hyporesponsiveness to different stimuli[29,30], perhaps explaining the increased susceptibility to early-onset sepsis observed in this patient group[31]. Similarly, those infants born to mothers with chorioamnionitis also exhibited significantly lower levels of FOXP3 expression in T-regs, suggesting reduced functionality[32].

Sex is known to have a significant bearing on neonatal outcomes with male preterm infants experiencing more adverse outcomes than females[33], and yet we saw very few significant differences in immune parameters between male and female infants. The few immune differences that were identified may, therefore, be functionally relevant. Both CD31 and its binding partner, CD38, were significantly lower in male CD4 T cells compared to their female counterparts, suggesting the whole pathway may be less efficient. Interestingly, CD38 has been shown to enhance T cell (and NK) activation by contributing to immune synapse formation[34,35] and CD31-deficient mice show

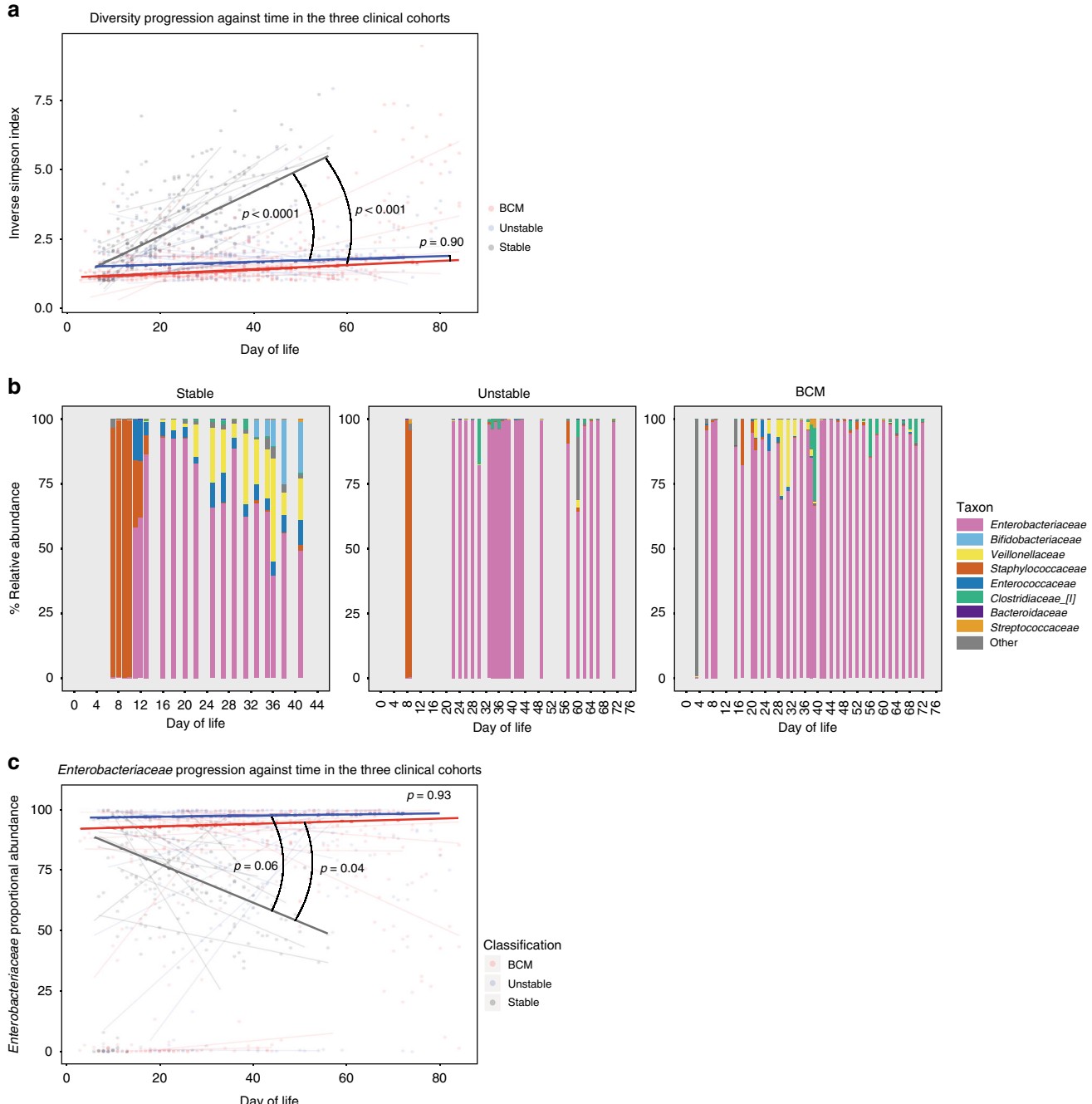

**Fig. 4 Microbiome development.** Faecal samples (713) were longitudinally collected from 34 babies across the duration of the study. Bacterial DNA was extracted, and the 16S rRNA gene (hypervariable region V4) was amplified, sequenced and compared against the sequence database to ascertain the relative abundance of bacterial taxa in samples. Participants were classified into three groups based on their clinical state: stable, unstable and those born to mothers with chorioamnionitis (BCM). **a** The progression of diversity against postnatal age across the three groups (stable—grey; unstable—blue; BCM —red). Dot plot represents individual sample diversity indices. Thin lines represent individual participants' regression coefficients through their diversity data points, as assessed by Theil–Sen estimator. Thick lines represent the median/mean of the individual trajectories within each clinical group. **b** Stacked bar charts of relative abundance of taxa in typical infants from the three clinical groups. Only taxa with >1% mean abundance across all samples are represented. **c** *Enterobacteriaceae* relative abundance against postnatal age across the three groups (stable—grey; unstable—blue; BCM—red). Dot plot represents individual sample *Enterobacteriaceae* proportions. Thin lines represent individual participants' regression coefficients through their *Enterobacteriaceae* proportions data points, as assessed by Theil–Sen estimator. Thick lines represent the median/mean of the individual trajectories within each clinical group. Source data are provided as a Source Data file.

enhanced susceptibility to LPS-induced endotoxic shock[36]. The only other factor that appeared to be sexually dimorphic was CD161 expression on CD8 T cells, which decreased in both male and female infants postnatally (albeit from an increased starting level in males). The expression of CD161 is thought to evoke a particular functional profile[37] irrelevant of the cell type. It is interesting, therefore, that expression was enhanced postnatally on NK cells and γδ T cells (where it correlated with IFN-γ production), but decreased on CD8 T cells. It is possible that some CD161$^+$ CD8$^+$ T cells represent mucosal-associated invariant T

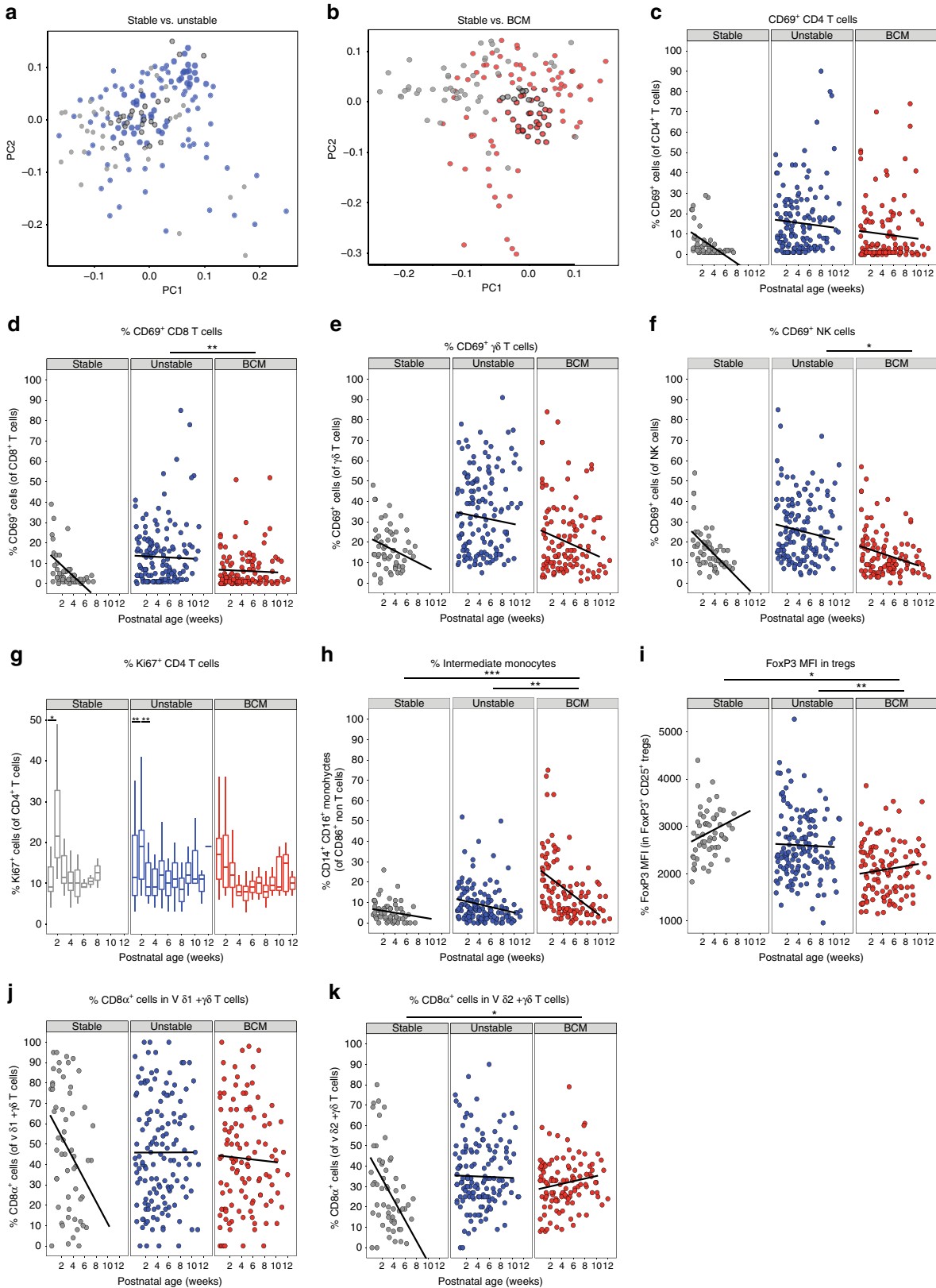

(MAIT) cells and thus the rapid decrease may be associated with migration out of the blood into the intestine. Interestingly, this decrease was not seen in the term infants, suggesting that, as opposed to the majority of parameters assessed, this may be dependent on GA, and gut homing of MAIT cells may take place prior to term birth.

The principal effector chemokine of neonatal CD4 T cells is CXCL8[10]. CXCL8 expression was highly divergent between infants and adults, and there was a strong correlation between CXCL8-producing CD4 and CD8 T cells in our babies, regardless of GA at birth. Levels of CXCL8-producing αβ T cells did not significantly decrease towards adult levels over the time period

**Fig. 5 Immune parameters altered by different pre- and/or postnatal exposures.** Longitudinal PBMC samples from 39 preterm babies were phenotyped for 186 different immune populations by flow cytometry following surface and intracellular staining. For cytokine detection, samples were activated in vitro with PI (4 h, in the presence of BFA) prior to staining. PCA of 186 immune parameters from longitudinal samples derived from: **a** stable (grey circles; $n = 10$ babies) and unstable (blue circles, $n = 16$) babies. **b** Stable (grey circles; $n = 10$) and BCM (red circles, $n = 13$) babies. In **c–f**, scatter plots depict frequencies of CD69+ cells in stable babies (left panel; grey circles, $n = 10$), unstable babies (middle panel; blue circles, $n = 16$) and BCM babies (right panel; red circles, $n = 13$) as a function of postnatal age: **c** CD69+ CD4 T cells, **d** CD69+ CD8 T cells, **e** CD69+ γδ T cells, **f** CD69+ NK cells are depicted from the three clinical groups. **g** Proliferation (as depicted by the boxplot showing frequency of Ki67 expression in CD4+ T cells) was elevated immediately post birth in infants born to mothers with chorioamnionitis, whereas stable (and to a lesser extent unstable) infants showed a proliferative burst at around 14 days of age. In this box-and-whisker plot, horizontal bars indicate the median, boxes indicate 25th to 75th percentile, and whiskers indicate 10th and 90th percentile. Compared to stable babies, unstable and BCM babies show significantly higher frequencies of intermediate monocytes (**h**) and significantly lower FOXP3 MFI in Tregs (**i**) and higher frequencies of CD8α-expressing γδ T cells over time in both **j** Vδ1+ γδ T cells and **k** Vδ2+ γδ T cells. Data shown are a pool of longitudinal samples from each clinical group where on average there are eight samples per baby in the stable cohort ($n = 10$) and nine samples per baby in the unstable cohort ($n = 16$) and the BCM cohort ($n = 13$). For all figures except **g**, \*\*\*$p < 0.001$, \*\*$p < 0.01$ and \*$p < 0.05$ as determined by linear mixed-effect modelling using the lmer package in R. For **g**, \*\*$p < 0.01$ and \*$p < 0.05$ as determined by a non-parametric Wilcoxon's matched-pairs signed-rank test. Source data are provided as a Source Data file.

studied, despite reductions in CXCL8-producing NK and γδ T cells. This notwithstanding, infants born in the context of chorioamnionitis or those with an unstable clinical course exhibited significantly lower CXCL8-producing CD4 and CD8 T cells when compared to stable infants. There may be several reasons for this observation. It is possible that these cells are somehow exhausted in the unstable cohorts. This is unlikely to be the case as the ability of the same cells to produce TNF or IL-2 was not reduced. Furthermore, both infection and inflammation may themselves be implicated. In this study, we show that babies born in the context of chorioamnionitis (inflammation) and those with repeated episodes of infection have lower CXCL8 responses but we do not know the mechanisms that link this association. The observation that CXCL8 and TNF production may be reciprocally associated with GA[14] and that unstable infants in this cohort were generally born at an earlier GA, may also be an explanation. However, amongst this cohort, there appeared to be no correlation between GA and the ability of CD4 T cells to produce CXCL8 or TNF around birth. Nevertheless, a reduced number of CXCL8-producing T cells was consistently observed in those infants with an unstable clinical course who were more likely to develop serious postnatal complications. This inability to mount adequate T cell CXCL8 responses in the first few days of life (for whatever reason) may therefore predict a poor outcome. This finding may have particular clinical relevance, both identifying CXCL8 as a potential therapeutic target but also as a biomarker to predict subsequent outcomes for babies born prematurely.

## Methods

**Ethical approval and funding.** Funding was granted by Barts Charity (Ref: 764/2306) for the undertaking of a study named 'Investigating Microbial Colonisation and Immune Conditioning in Preterm Babies'. Ethical and regulatory approvals were granted by the London (Chelsea) Research Ethics Committee (Ref: 15/LO/1924) and the local Research and Development department at Homerton University Hospital NHS Foundation Trust. Term infant blood samples were collected with informed consent as part of the Infectious Disease Biobank, ethical approval granted by South Central—Hampshire B Research Ethics Committee (Ref: 09/H0504/39) from infants undergoing cardiac surgery. Written informed consent was obtained in every case.

**Participant enrolment.** Participants were recruited from the neonatal intensive care unit at Homerton University Hospital between January 2016 and December 2017. Babies born between 23 weeks and 4 days and 31 weeks and 6 days of gestation admitted to the NICU before 72 h postnatal age, including those born at other hospitals, were included in this study. Of those participants eligible for recruitment, 22% were not approached as they were thought to have no realistic chance of survival, had known exposure to either HIV or hepatitis B infection or due to other concerns, including capacity to provide consent and availability of the research team. Written informed consent was obtained from parents before 72 h of age, and of the parents approached, 62% agreed to participate.

**Sample collection and storage.** Blood samples (0.5 ml whole blood) from preterm infants were collected on a weekly basis, with additional samples taken at times of suspected infection, up to cessation of the infants' participation. Blood samples (0.5 ml whole blood) from term infants were collected prior to cardiac surgery. All samples were collected opportunistically during routine sampling for clinical indications (as mandated by the Ethics Review Committee) and were stored at room temperature (on average for 12 h and up to a maximum of 30 h) prior to Ficoll separation of peripheral blood mononuclear cells (PBMCs), and were frozen in Cryostore (Sigma) and stored in liquid nitrogen. Faecal samples were collected on a daily basis (when available) from birth up to cessation of the infants' participation in the study (i.e. discharge or transfer). Samples were immediately stored at 4 °C for <72h, before transferring to long-term storage at −80 °C.

**Clinical data.** Detailed demographic, antenatal and daily postnatal data were collected prospectively on all preterm infants. Once each subject completed their hospital stay, their clinical course was reviewed by the clinical team and the participants classified a priori into one of three clinical cohorts:

- All infants born to mothers with a history of clinical or histologically confirmed chorioamnionitis
- Stable infants—those infants receiving a single course of antibiotics only in the immediate postnatal period, with no episodes of suspected late-onset sepsis or antibiotic exposure thereafter
- Unstable infants—those infants who did not meet the stable criteria, having received additional courses of antibiotics for suspected late-onset sepsis or other infection concerns including NEC.

The collaborating immunology team was not aware of the clinical status of babies when blood samples were collected and/or analysed. Once the laboratory analyses were completed, the participants' clinical courses were provided by the clinical team. No clinical data (other than GA at birth, postnatal age (<6 months) and the fact they were undergoing cardiac surgery for non-immune-related conditions) were collected on the term infants.

**Cell stimulation.** PBMCs were stimulated with phorbol myristate acetate and ionomycin as described below. Cells were stimulated in RPMI-1640 medium (Invitrogen) containing 10% (vol/vol) FCS (StemCell Technologies), 2 mM L-glutamine (Sigma), 100 U penicillin and 100 µg/ml streptomycin (Invitrogen) (complete medium, CM) containing 10 ng/ml phorbol 12-myristate 13-acetate (Sigma) and 1 µg/ml ionomycin (Sigma) for 4 h at 37 °C, 5% $CO_2$ in the presence of 20 µg/ml brefeldin A (Sigma). For spontaneous cytokine production, cells were incubated in CM in the presence of brefeldin A only.

**Flow cytometry.** All samples from individual babies were analysed simultaneously to avoid introducing any unnecessary experimental variation. Single-cell suspensions were prepared in FACS buffer (phosphate-buffered saline plus 2.5% (vol/vol) foetal calf serum and 2 mM EDTA) and then split into nine aliquots for staining with nine different 11-colour flow cytometry panels. To one of the aliquots, 50 µl of Countbright Absolute counting beads (catalogue no. C36950, Life Technologies) was added. Initially, cells were incubated on ice for 15 min with Zombie NIR Fixable Viability Kit (BioLegend) to allow live/dead discrimination. For panels containing Vδ1, the surface staining antibody was added along with the live/dead discrimination stain. Cells were then surface stained on ice for 30 min with antibodies listed in Supplementary Table 1. Following surface staining, cells were fixed (Cell Fix, BD) and then permeabilised using Perm buffer (BioLegend). For panels containing transcription factor FoxP3, cells were fixed and permeablilised using fixation and permeabilising reagents from Invitrogen's Intracellular Fixation & Permeabilisation Buffer Set (catalogue no. 88-8824-00) according to the manufacturer's instructions. Following permeabilisation, cells were then intracellulary stained for

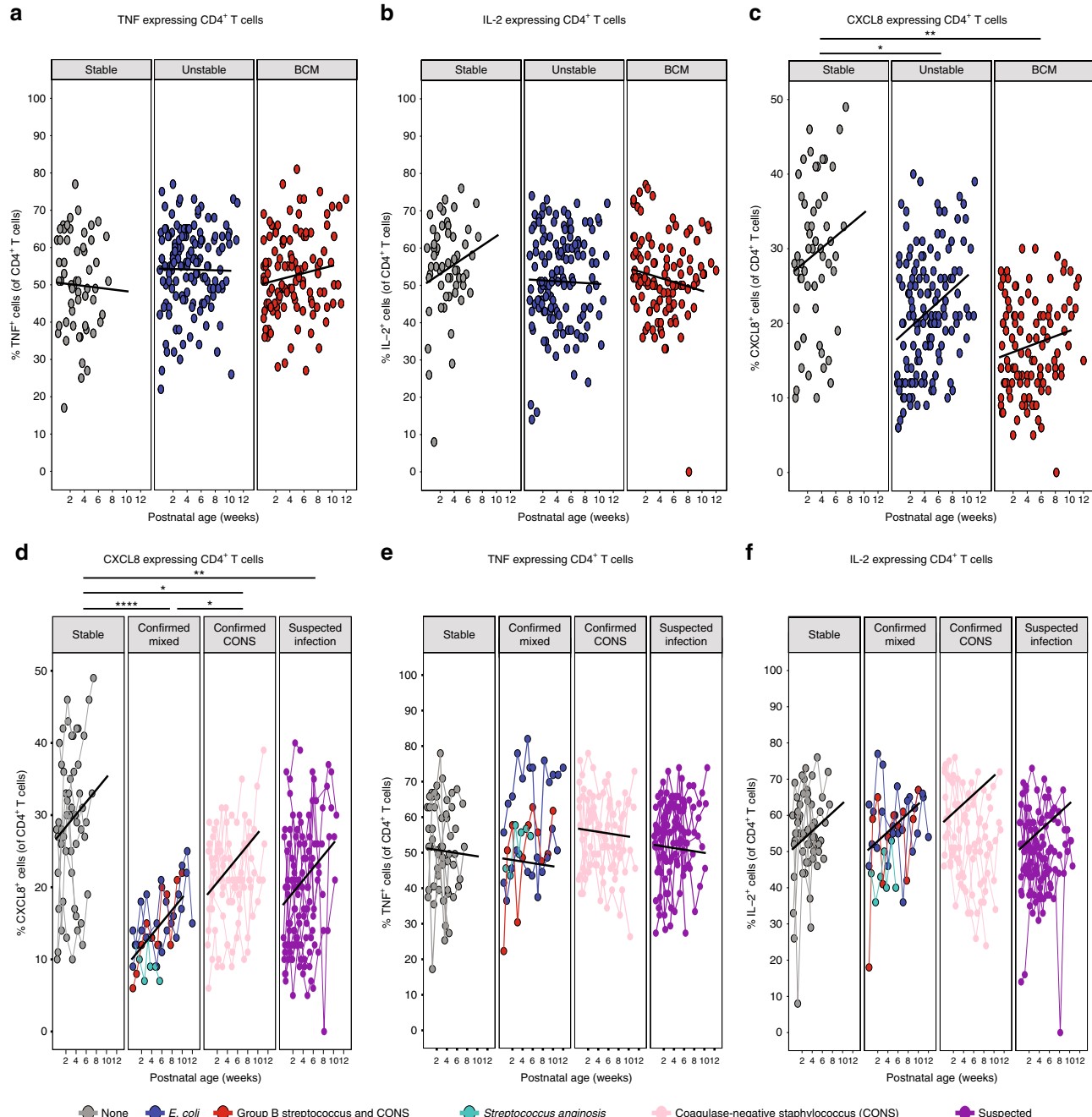

**Fig. 6 Unstable infants have significantly reduced CXCL8-producing T cells compared to stable babies.** Longitudinal PBMC samples from 39 preterm babies were activated in vitro with PI (4 h, in the presence of BFA) and expression of CXCL8, TNF and IL-2 assessed by flow cytometry. Scatter plots showing frequencies of **a** TNF, **b** IL-2 and **c** CXCL8 in stable babies (left panel; grey circles, $n = 10$), unstable babies (middle panel; blue circles, $n = 16$) and BCM babies (right panel; red circles, $n = 13$) as a function of postnatal age. In **a**–**c**, data shown are a pool of longitudinal samples from each clinical group where each circle represents a longitudinal sample from an individual baby and on average there are six samples per baby in the stable cohort and nine samples per baby in the unstable cohort and the BCM cohort. Blood culture-positive bacteraemias were separated into those from coagulase-negative staphylococci (CoNS) and other bacteria as CoNS infections are generally considered to be less severe with lower mortality rates. Scatter plots showing the frequencies of **d** CXCL8, **e** TNF and **f** IL-2 in stable babies (far left panel; grey circles, $n = 10$ babies with a mean of eight longitudinal samples per baby), babies with microbiologically confirmed infections (second panel; $n = 4$ with a mean of 11 longitudinal samples per baby), babies with coagulase-negative staphylococcal infections (CoNS) (third panel, $n = 9$ with a mean of 10 longitudinal samples per baby) and babies with suspected infections (far right panel; $n = 16$ with mean of seven longitudinal samples per baby). The colour of the circles depicts the type of infection as described in the figure. In **d**–**f**, each circle represents a longitudinal sample from an individual baby and linked circles represent longitudinal samples from the same baby. ***$p < 0.001$, **$p < 0.01$ and *$p < 0.05$ as determined by linear mixed-effect modelling using the lmer package in R. Source data are provided as a Source Data file.

cytokines and transcription factor FOXP3 with antibodies listed in Supplementary Table 1. All samples were run on a four laser BD LSRFortessa X20 flow cytometer. To minimise the impact of temporal variation in instrument sensitivity, cytometer performance was monitored using CS+T beads (BD) and experiments were run using applications settings. Data were analysed using the FlowJo software. All gating strategies utilised are shown in Supplementary Figs. 10–15.

**Laboratory methodology for faecal DNA extraction**. Faecal samples were selected for analysis on a pragmatic basis to provide regularity of assessment throughout the infants' admissions, with increased resolution at times of clinical instability, and subject to availability—samples were selected approximately every 3 days (where possible), with additional samples taken before/during/after courses of antibiotics, feeding type changes, ventilation changes, and blood transfusions, to capture possible associated microbiome changes.

Faecal bacterial DNA was extracted using the DNeasy® PowerSoil® Kit (Qiagen, Netherlands), following treatment with propidium monoazide, to prevent downstream amplification of contaminant or non-viable free bacterial DNA. The V4 hypervariable region of the 16S rRNA gene was amplified using PCR with a dual-indexing strategy, and sequencing of the subsequent amplicons was performed using Illumina MiSeq technology (2 × 250 bp flow cell for paired-end sequencing with 5% PhiX DNA).

**Faecal microbiome analysis**. Sequencing data were processed using the DADA2 pipeline and all analyses and graphics were produced by means of R packages (DADA2, decontam, tidyr, dplyr, ggplot2 and mblm)[38–44]. Microbiome analyses were performed on 10 stable (100% of eligible participants), 13 unstable (81%) and 11 chorioamnionitis (84%) infants.

Biometrics of microbiome structure were used to describe the development of bacterial communities within the cohort: for the description of constituent taxa, analysis was performed at the family level, as this represented the lowest taxonomic level with the highest level of read identification (99.9% of amplicons were identified at family level; only 74% were identified at genus level); only those taxa with a mean relative abundance of >1% across all samples were included in the analyses. Diversity was estimated using the inverse Simpson statistic, and was derived from all taxa (regardless of relative abundance). The longitudinal progression of the relative abundances of individual family-level taxa were described.

**Statistical analysis**. Difference between neonatal and adult immune profiles: To understand the degree to which the immune profile of neonates differed from that of adults, we performed a two-tailed non-parametric Wilcoxon's rank-sum test with a Benjamini–Hochberg correction for multiple comparisons between adult samples ($n = 9$) and the first time point neonatal blood sample for each individual ($n = 33$).

**Longitudinally changing data sets**. To identify immune parameters that changed significantly over time, longitudinal models of the cell-subset frequencies and total numbers at the group level were constructed. Group level slopes were estimated using mixed linear models of the cellular frequencies/numbers versus postnatal age, with a random intercept and slope at an individual level (LME4 R package)[45]. The regression lines were weighted by the number of samples per baby, which varied between infants. To determine whether a given immune parameter changed significantly between one time point and another, a non-parametric Wilcoxon's matched-pairs signed-rank test was used.

To assess the longitudinal development of microbial communities over time, regression coefficients (Theil–Sen non-parametric regression coefficient) for diversity and *Enterobacteriaceae* relative abundance against time were calculated as summary measures to describe the longitudinal progression of individual participants. These summary measures were then used to derive group level summary statistics (i.e. median), and to allow statistical comparison between the groups (by weighted Mann–Whitney *U* test). Summary measures were weighted to account for the number of originating data points (corresponding to numbers of samples) from which the individual measures were derived. Only the family *Enterobacteriaceae* was examined in this manner, as it was the only taxon widely distributed across the data set in significant relative abundances (>40% mean abundance in >90% of all participants) to allow comparison between the three clinical groups. Longitudinal progression of individual taxa in the three clinical cohorts is represented by a plot of relative abundance of individual taxa at each time point. In recognition of the bimodal distribution of some taxa, we highlighted the infants who were essentially not colonised with specific taxa at any time point (maximum relative abundance <1%).

To understand the relationships that exist between immune parameters and microbiome parameters, intra-individual Spearman's correlation was performed. To account for variation in the sampling schedule between blood and stool, we binned the data into 5-day time windows to match blood and contemporaneous stool samples. In the event of more than one sample per individual occurring in any given bin, mean values were calculated for every parameter measured. Spearman's correlation analyses were performed across all parameters barring those directly linked to one another owing to the nested nature of flow cytometry

analysis (e.g. in the case of CD4 and CD8 as a percentage of total αβ T cells, only CD4 T cells are represented in the analysis). A mean value was derived for the correlation coefficient between all parameters across all individuals. For a relationship to be considered significant, it would have to consistently yield a $p$ value of <0.2 in every individual tested, the aggregate of this across all individuals equates to a $p$ value threshold of $p = 0.008$. Stool passage was unpredictable among babies recruited to the study, so not all infants had sufficient samples to perform a 100% match between immunological and microbiome parameters.

**Reporting summary**. Further information on research design is available in the Nature Research Reporting Summary linked to this article.

## Data availability
The flow cytometry data (FCS files) has been deposited in Flow Repository (http://flowrepository.org/id/FR-FCM-Z2FJ). The 16S rRNA gene sequence data has been deposited onto NCBI BioProject accession reference PRJNA605031.

## Code availability
We did not use any custom code. The code packages used are listed here: DADA2, decontam, ggplot2, mblm, dplyr, tidyr, lme4, Viridis, ViridisLite, data.table, Rcolorbrewer, ggrepel, ggfortify, ggbeeswarm and reshape2, gridExtra, Hmisc, Rtsne and corrplot.

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

## Acknowledgements

We thank S. Evans for help with statistical analyses; I. Jogee for illustrations; A. Das and F. Kyle for some sample processing; C. Austin for term infant samples; A. Hayday (KCL) for critical review of the manuscript. P.F., R.H., F.S., K.A., K.C. and M.R.M. were all supported by a strategic research grant from Barts Charity (Grant 764/2306); S.K. and D.G. by Cancer Research UK (Grant A20730); A.L. by the Wellcome Trust (Grant 100156/Z/12/Z to A. Hayday), and W.G.W. by Queen Mary University of London.

## Author contributions

P.F., R.H. and F.S. did all the patient and sample recruitment; S.K. did blood processing, all the flow cytometry experiments and data analysis; R.H. and K.A. did DNA extraction from stool, next-generation sequencing and R.H. analysed the microbiome data; A.L. helped design and optimise flow cytometry panels and analysed large data sets; W.G.W., M.R.M. and K.C. provided critical review of the data; P.F. set up the study, evaluated the clinical data and co-wrote the manuscript; D.G. designed the immunology study, evaluated the results and co-wrote the manuscript.

## Competing interests

The authors declare no competing interests.
