## [Peer Review File · Nature Communications]

Reviewers' comments:

Reviewer #1 (Pediatrics, systems immunology)(Remarks to the Author):

Review

Kamdar et al analyze immune cell frequencies and composition in preterm children in relation to clinical features and the microbiome. The topic is of importance, the data seems very interesting, but the data analyses performed are rudimentary and the study feels like a missed opportunity. Several of the main conclusions could have alternative explanations not explored by the authors. Also, neither the data, nor the analysis code seem to be deposited in any public domains, preventing others from reanalyzing the data and me from properly reviewing the work.

Specific points:

1. In Fig 1, newborn immune features are compared to adults, but adults show a wide range of variation, as expected from previous work (Patin et al, Nat Imm 2018, Roederer et al, Cell 2015, Brodin et al, Cell 2015 etc). What is the relevance of comparing to adults? Parents would be a more reasonable comparison here given the genetic and environmental relatedness to the newborns in the study. If parents are not available, comparing distributions in newborns to those of adults using Kolmogorov-Smirnoff tests or similar would be more appropriate than the current approach.
2. Cell frequencies seem to be treated as independent variables throughout the study, when they are in fact highly dependent on each other. As an example, is the B-cell increase due to an actual increase in B-cells or a decrease in some other subsets? In Figure 2 correlations are calculated but these are difficult to interpret given the inherent relationships between features
3. The microbiome analysis is also deficient in that global compositional features are not considered and instead only one single family of bacteria is analyzed. Previous studies have shown a patterned progression in preterm gut microbiomes (La Rosa et al, PNAS 2014). How do the authors reconcile the lack of such a patterned progression seen here?
4. Some of the microbiome findings seen are consistent with previous reports and could be explained by mode of delivery (High Staph early in life in C-section delivered children). This should be discussed in the manuscript.
5. The lack of correlations between microbes and immune features could be due to the failure to consider the pairing of samples from specific individuals. It is my understanding that all immune samples are considered separately, and all microbiome samples considered separately, even though these are in fact linked series of samples from the same individuals. Taking such relationships into account is mandatory in my opinion.

All in all, the authors clearly have a very interesting dataset generated but have failed to capitalize on it and most of the current conclusions are uncertain and not robustly supported by the data. I suggest a complete rework of the analyses, taking feature and sample relationships into account is required to draw meaningful conclusions.

Reviewer #2 (Newborn immunology, microbiome)(Remarks to the Author):

There are several concerns:

Main concerns:

- Lack of term infant comparison - clearly during the neonatal period, this would be the 'gold standard' comparison, not adults as done in the manuscript. Further, with adults there likely will have only been one measurement, ie no longitudinal assessment. Even in term infants massive changes occur in the first few weeks. Although they may be more dramatic in preterm (as this manuscript posits),

comparing them to term newborns would be the appropriate comparator. If this has not been done, please explain why.

- All results in graphs are given in age postnatal weeks, not days. So I cannot quite follow how they can state 'at birth' ? Timing of sampling is hugely important.
- Storage time is given as an average of 12h and up to 30h at room temperature. Not only is such long storage time known to be a problem (introduces leukocyte adhesion to plastic, degranulation, changes in cytokines, activation marker expression etc.), it also is not clear as to how much difference there was between samples reg. storage. I would only expect much variation for samples taken at the time of sepsis, not the routine weekly samples (as they knew they were coming). Such systematic difference in storage time may have skewed the results.
- The classification of their study subjects is utterly incomplete. For example, it is unclear what the chorioamnionitis status of the 'stable' infants were ? It should be similar in distribution as in the unstable infants and should be taken into account for these infants too, since if that were a critical driver of immune responses, it should be present there too. Furthermore, how is this impacted by the vastly different GA between the groups? At 30w (stable infants) the expected incidence of chorio is <10% whereas at 25w (unstable with chorio group) it more likely was 40-50%. Further lacking is characterization of the degree of placental inflammation (clearly of massive importance with Chorio), including specifically involvement of maternal, fetal or both placental layers. Were Cultures taken and positive?
- Statistical analysis: the description of immune parameter analysis is a bit brief and doesn't clearly state which confounders were assessed and what parameters were adjusted for
- Postmenstrual age rather than postnatal age may be better to compare the effects of GA at birth (which is lost if only looking at postnatal time periods).
- Patient demographics: a full flow chart would be advisable; ie how many patients approached, declined, excluded, died etc. Surprising to see no infant with NEC, were these excluded (and yet the stable states outcome GI pathology) ? And again: what is the chorioamnionitis status of the 'stable' babies ? Comparison between stable and unstable groups is mute - authors state themselves these were more mature, bigger and all born via CS, so not really a control group. At least these parameters need to be adjusted for in analysis. They may simply have been more stable because of these factors. Very likely even looking at the mean GA of the groups might reveal confounders (of note: there seems to be no comparison of e.g. 30.5 vs 28.7 vs 25.4w., which we know are entirely different clinically, even without any tests. This may have massive impact, as either the younger babies die more often (hence request for flow chart).
- Would also strongly suggest to add range not just interquartiles, as this will provide a more realistic view of the data spread
- Microbiome analysis: should specify this is ONLY stool analysis, not THE microbiome, we do not know yet what happens correspondingly at other body sites.
- I am not sure how the propidium iodide addition would prevent actual bacterial contamination. In principle, PI binds to exposed DNA. It should bind any free DNA and it would enter non-viable cells and bind their DNA. Subsequently, only DNA from initially viable cells is supposed to be amplifiable by PCR due to the PI intercalation. However, as the remaining free PI is washed out, this would not prevent contamination from a) viable cells, b) nor would it prevent contamination from DNA in PCR reagents which would be added after the PI treatment.
- They describe differences in diversity and Enterobacteriaceae progression between the stable/unstable groups. Considering demographic differences between these groups, has this been adjusted for GA? Type of milk feeds? Was there probiotic supplementation? What about mode of delivery? Etc.

Minor points:

- Use of PMA/ionomycin as sole immune stimulus is concerning. While it often is considered one of the 'standard' stimuli, it is rather strong and - importantly - entirely unphysiological and may, in my

opinion, miss more subtle yet important differences. Why were more specific stimuli (TLR ligands or TCR stimuli) not used?

- The various statements reg. immune functions 'at birth' are factually incorrect. Consent was taken postnatally within 72h, not antenatally. Also, I don't think they looked at cord blood.
- Line 394. This is not unexpected once taking GA at birth into account.
- Faecal sampling 2.3 states daily; 2.7 states every 3 days ? What is it?
- With this frequency of sampling a huge number of samples should be available. Could they indicate how many in total per infant ? Were they all analysed? If not, why not?
- How can it be they only have microbiome analysis in 2/3 of the unstable infants? Does this still support the claim that 'microbiome' is unrelated to immune development ?
- Fig 3A - very hard to make out the important lines, consider change in colour scheme ?
- Line 343 'most severe infections' - not defined
- Demographics - given the relevant differences between groups I suspect the rates of antenatal antibiotic therapy were very different ? This simply must be stated.

NCOMMS-19-24100-T. Response to reviewers

Please find the responses to all the comments made by the reviewers below.

Reviewer #1

Kamdar et al analyze immune cell frequencies and composition in preterm children in relation to clinical features and the microbiome. The topic is of importance, the data seems very interesting, but the data analyses performed are rudimentary and the study feels like a missed opportunity. Several of the main conclusions could have alternative explanations not explored by the authors. Also, neither the data, nor the analysis code seem to be deposited in any public domains, preventing others from reanalyzing the data and me from properly reviewing the work.

We also believe the data are very interesting and we apologise that the methods of data analysis were not clear and hence appeared rudimentary. This was not the case and the exact analysis has been detailed below and in the revised paper. The data will be deposited on the open science framework (OSF) <https://osf.io> and the FCS files will be deposited on the flow repository (<https://flowrepository.org>) post acceptance and prior to publication.

Specific points:

1. In Fig 1, newborn immune features are compared to adults, but adults show a wide range of variation, as expected from previous work (Patin et al, Nat Imm 2018, Roederer et al, Cell 2015, Brodin et al, Cell 2015 etc). What is the relevance of comparing to adults?

Whilst we agree that adults are very variable and we would expect variation between adults and infants, the comparison to adults was merely to highlight which parameters are the most distinct in preterm around the time of birth-hence only a supplementary figure. As our infants were followed longitudinally, we were then able to see which parameters moved towards the levels in adults in the 3 month period which we do feel is a very valid comparison (Fig 2a). This was particularly interesting as some parameters did reach near adult levels whilst others did not which we comment on.

Parents would be a more reasonable comparison here given the genetic and environmental relatedness to the newborns in the study. If parents are not available, comparing distributions in newborns to those of adults using Kolmogorov-Smirnoff tests or similar would be more appropriate than the current approach.

Unfortunately, parents were not available. With regards the statistics, we thank the reviewer and acknowledge that a straightforward t test is probably not the most appropriate test in this instance. We have reanalysed the data using a Wilcoxon rank sum test with a Benjamini-Hochberg correction for multiple comparisons which is more robust given the potential for the data to be non-normally distributed. Whilst the magnitude of the

p values has decreased as a result, the overall interpretation of the data has remained unchanged. The revised Supplementary figure 1 is added.

2. Cell frequencies seem to be treated as independent variables throughout the study, when they are in fact highly dependent on each other. As an example, is the B-cell increase due to an actual increase in B-cells or a decrease in some other subsets?

We are aware that many parameters are linked, and where possible we did show actual increases in cell numbers to avoid this (B cells are a good example as questioned by the reviewer, an increase in actual B cell numbers was shown in Supplementary figure 2A). Actual cell numbers are also shown for NKG2D expressing NK cells (Suppl Fig 2C).

In Figure 2 correlations are calculated but these are difficult to interpret given the inherent relationships between features

As we are aware of the reviewer's point regarding variables depending on each other, in this figure we only plotted one parameter (of a correlated pair) if it was inherently related to another parameter (for example, $\gamma\delta$ T cells lacking CD161 expressing IFN- γ were retained whereas the linked IFN- γ negative, CD161 negative $\gamma\delta$ T cell population was removed from the analysis); this was done for all linked populations. Apologies this was unclear, this has now been altered in the text for clarity.

3. The microbiome analysis is also deficient in that global compositional features are not considered and instead only one single family of bacteria is analyzed. Previous studies have shown a patterned progression in preterm gut microbiomes (La Rosa et al, PNAS 2014). How do the authors reconcile the lack of such a patterned progression seen here?

We are aware of the La Rosa paper and its contribution to understanding longitudinal microbiome trends. Perhaps the most crucial differences between their microbiome analyses and ours are that their groups are composed of a heterogeneous mixture of stable and unstable babies, with variable (and from assessment of their study demographics) extensive use of antibiotics. Consequently, the trends they have highlighted are a composite of those of stable and unstable babies. In our cohort, these have been separated to account for antibiotic exposure and therefore minimise the effects of such a potentially powerful modulator of the gut microbiome, something deficient within the wide body of literature pertaining to the preterm gut. They also describe their findings at a class taxonomic level, whereas we have described it at a more detailed family level which makes comparison difficult. The reason that only Enterobacteriaceae progression is described is that examination of individual subjects' longitudinal trends demonstrated this to be the only taxon which was consistently described across all subjects in our study. The remainder of taxa showed a multimodal distribution of colonisation, meaning that the use of summary statistics (e.g. mean/median) in describing them would have been misleading." We do present the data for select taxa (Enterococcaceae, Bifidobacteriaceae, Staphylococcaceae and Veillonellaceae), but summary measures and statistical analyses on these data would have been inappropriate.

4. Some of the microbiome findings seen are consistent with previous reports and could be explained by mode of delivery (High Staph early in life in C-section delivered children). This should be discussed in the manuscript.

The observed association of high Staphylococcaceae abundance in early life in the stable group (uniformly born by operative delivery) is consistent with the previously noted association of staphylococci colonisation after caesarean section (as opposed to natural birth) (Itani et al, 2017, Anaerobe; Shao et al, 2019, Nature). These points are now raised in the manuscript and additional references added.

5. The lack of correlations between microbes and immune features could be due to the failure to consider the pairing of samples from specific individuals. It is my understanding that all immune samples are considered separately, and all microbiome samples considered separately, even though these are in fact linked series of samples from the same individuals. Taking such relationships into account is mandatory in my opinion.

We apologise our analysis was unclear and hence we have elaborated in the manuscript text. For an individual baby, immune measures were mapped to contemporaneous microbiome analysis and a correlation established. The correlation values were then aggregated across all individuals to form the figure. The immune and microbiome samples were NOT considered separately but as linked data within individual babies. The relationships were then pooled to form the figures.

All in all, the authors clearly have a very interesting dataset generated but have failed to capitalize on it and most of the current conclusions are uncertain and not robustly supported by the data. I suggest a complete rework of the analyses, taking feature and sample relationships into account is required to draw meaningful conclusions.

Thank you for the detailed review of our manuscript and data and for the suggestions you have made. We hope we have addressed these in the above responses and highlighted that we did analyse our data appropriately taking feature and sample relationships into account and have now made this much clearer with apologies that we did not do this sufficiently well initially.

Reviewer #2 (Newborn immunology, microbiome)(Remarks to the Author):

There are several concerns:
Main concerns:

- Lack of term infant comparison - clearly during the neonatal period, this would be the 'gold standard' comparison, not adults as done in the manuscript. Further, with adults there likely will have only been one measurement, ie no longitudinal assessment. Even in term infants massive changes occur in the first few weeks. Although they may be more dramatic in

preterm (as this manuscript posits), comparing them to term newborns would be the appropriate comparator. If this has not been done, please explain why.

As we mention above, comparison to adult was merely to highlight any progression in immune parameters over time towards the levels observed in adults. Within the UK (and likely further afield), it would be difficult to convince an ethics review board to allow repeated sampling in healthy term babies, which was why this was not done as the 'gold' standard comparator group. Preterm infants in hospital are subject to frequent blood tests to monitor their clinical progress. All blood samples taken for this study were taken at the time of routine blood taking for clinical purposes. Indeed, I have contacted a former Chair of a clinical research ethics committee and this was confirmed. While research on healthy (adult) volunteers may be justified by being 'in the public interest', this could not be invoked for healthy babies who very rarely suffer bacterial sepsis. Furthermore, if such a study was approved by a Research Ethics Committee, and indeed the MRC guidelines suggest it is possible, it would be unlikely that parents would agree to routine blood taking for a purpose unrelated to the health of their baby. In addition, the postnatal course for a healthy term baby is not the same as that for a baby born at <32 weeks i.e. it would not be expected to be interrupted by repeated evaluations for suspected infection. Any comparison would therefore be logistically and ethically challenging and of possibly limited value. However, in an attempt to investigate whether the changes observed in our cohort are also mirrored in a term baby cohort, we have analysed different term babies sampled at different ages in the first few months of life. These samples were blood samples from infants prior to cardiac surgery in a separate study on thymus development. As these babies were not followed longitudinally there was, as expected, significant heterogeneity in the different cell subsets, to a similar extent to that seen in the preterm cohort when looking between different babies which all had their own individual profiles (Fig 3C). However, when we looked at several parameters which changed in our preterm cohort over time, interestingly, term babies seem to gain functionality at a similar rate suggesting similar postnatal adaptation in both cohorts. Exceptions to this are discussed. This data has been added as a supplementary data figure (Supplementary Figure 4).

- All results in graphs are given in age postnatal weeks, not days. So I cannot quite follow how they can state 'at birth' ? Timing of sampling is hugely important.

We apologise about our use of 'at birth'. The reviewer is correct as this is inaccurate and we have changed this throughout the manuscript, 'at birth' has been removed throughout to avoid any inaccuracies. We do not believe, however that changing the labelling on the graphs into days would have any additional benefit, other than making the axis labels harder to read.

- Storage time is given as an average of 12h and up to 30h at room temperature. Not only is such long storage time known to be a problem (introduces leukocyte adhesion to plastic, degranulation, changes in cytokines, activation marker expression etc.), it also is not clear as to how much difference there was between samples reg. storage. I would only expect much

variation for samples taken at the time of sepsis, not the routine weekly samples (as they knew they were coming). Such systematic difference in storage time may have skewed the results.

Unfortunately, as our ethics did not allow collection specifically for this study, even weekly samples were taken only when routine bloods were being taken (often during the night shift) and hence timing was not always convenient. The samples were collected as simultaneously as our ethical permission and clinical opportunity permitted. We fully agree that some populations are lost with extended time frame between sample collection and processing. Indeed, we ran some initial tests on the same blood samples processed at different times to identify such populations. Actually, there were very few of the populations of interest that were affected. Intermediate monocytes were such a population and hence we have added into the manuscript how levels of these are probably underrepresented. As we were wary of this, we did careful observations for any populations that may have appeared decreased etc and checked the timing of the sample. We noted both collection and processing time on each sample so this could be monitored. We excluded any samples that were processed more than 30 hrs after collection from the analyses.

- The classification of their study subjects is utterly incomplete. For example, it is unclear what the chorioamnionitis status of the 'stable' infants were? It should be similar in distribution as in the unstable infants and should be taken into account for these infants too, since if that were a critical driver of immune responses, it should be present there too.

We apologise that this was unclear. We have clarified this throughout the paper. For ease of understanding the groups are classified as below. There are three groups designated post collection and assessment of clinical course. Those born with histologically confirmed or clinical chorioamnionitis were designated as 1 group. Those infants born in the absence of any evidence of chorioamnionitis were then further subdivided into clinically stable (no antibiotic exposure beyond the first week after birth) and clinically unstable (several repeated episodes of antibiotic treatment in association with suspected infection). With the exception of just one baby, all the babies in the chorioamnionitis group were also clinically unstable requiring several episodes of antibiotic treatment in association with suspected infection. We made this separation in order to assess and indeed overcome the potential confounder presented by the presence of chorioamnionitis.

Furthermore, how is this impacted by the vastly different GA between the groups? At 30w (stable infants) the expected incidence of chorio is <10% whereas at 25w (unstable with chorio group) it more likely was 40-50%. Further lacking is characterization of the degree of placental inflammation (clearly of massive importance with Chorio), including specifically involvement of maternal, fetal or both placental layers. Were Cultures taken and positive?

All the stable babies had no evidence of chorioamnionitis as described above and now made clearer in the manuscript– I would not expect the distribution of chorioamnionitis diagnosis to be the same in the post-hoc allocated stable and unstable groups, as a history

of chorioamnionitis is an independent risk factor for complications in the preterm course, thus making them more likely to be retrospectively classified as ‘unstable’. Indeed, as now made clear, all but one of the infants born in the context of chorioamnionitis, had an unstable clinical course. I agree that the differing corrected gestational age at birth between the stable and unstable groups is a potentially significant confounder which we now make more reference to in the manuscript.

We provide a table of the chorioamnionitis descriptions for babies in the chorioamnionitis group for the reviewer. Placental swabs were not taken. We do not feel that this detail would be of significant interest to the average reader but can, of course, add this into the manuscript if the editor felt it was of additional value

Subject	Chorioamnionitis Description	Fetal Inflammation Description
18	Severe acute chorioamnionitis	Mild funisitis
39	Acute chorioamnionitis	Umbilical vasculitis (mild-moderate FIRS)
40	Chorioamnionitis	Focal chorionic vasculitis (mild FIRS)
43	Chorioamnionitis	Umbilical vasculitis (moderate FIRS)
44	Acute chorioamnionitis	Early fetal phlebitis
49	Acute chorioamnionitis (severe)	Funisitis (moderate FIRS)
51	Not sent	
59	Acute chorioamnionitis	Vasculitis and funisitis (advanced grade 1 FIRS)
60	Acute chorioamnionitis	No FIRS
66	Acute chorioamnionitis	Vasculitis and funisitis (advanced grade 1 FIRS)
70	Acute chorioamnionitis	Vasculitis and funisitis (intermediate grade 1 FIRS)
76	Acute chorioamnionitis	Vasculitis (mild-moderate FIRS at intermediate stage)
101	Acute chorioamnionitis and	Umbilical phlebitis indicating a maternal and fetal response

- Statistical analysis: the description of immune parameter analysis is a bit brief and doesn't clearly state which confounders were assessed and what parameters were adjusted for

In the microbiome analyses, the biometrics are derived from intra-subject trajectories (independent of gestational age), so this would not have made a difference. The description of the immune parameter analysis has been extended.

- Postmenstrual age rather than postnatal age may be better to compare the effects of GA at birth (which is lost if only looking at postnatal time periods).

We have also performed all the analyses using post menstrual age and the graphs are more or less identical to those plotted by postnatal age-this suggests that the changes are driven by environmental effects post birth, this is now fully explained in the text

- Patient demographics: a full flow chart would be advisable; ie how many patients approached, declined, excluded, died etc.

Of the subjects eligible for recruitment, 22% were not approached for consent due to concerns regarding infant short term survival or due to other concerns including capacity to provide consent and availability of the research team. Of the parents approached, 62% agreed that their infants could be enrolled. Stool samples were not collected in just 6% of these infants. This information has been added into the paper and also the figure legend of the protocol (Fig 1). Of the babies represented in this data set (n=39), none of the babies died.

Surprising to see no infant with NEC, were these excluded (and yet the stable states outcome GI pathology) ?

Of the five babies in the unstable group three were diagnosed with NEC stage II. These data are updated in the amended table.

And again: what is the chorioamnionitis status of the 'stable' babies ?

This is addressed above. Any infant with chorioamnionitis is in one group. All but one infant in the chorioamnionitis group had an unstable clinical course.

Comparison between stable and unstable groups is mute - authors state themselves these were more mature, bigger and all born via CS, so not really a control group. At least these parameters need to be adjusted for in analysis. They may simply have been more stable because of these factors. Very likely even looking at the mean GA of the groups might reveal confounders (of note: there seems to be no comparison of e.g. 30.5 vs 28.7 vs 25.4w., which we know are entirely different clinically, even without any tests. This may have massive impact, as either the younger babies die more often (hence request for flow chart).

We agree that the differences in clinical stability may be affected by the differing gestational ages in the cohorts. We would not try to suggest otherwise and indeed, did not suggest that the stable infants are a 'control' group. Comparisons between the groups did highlight differences which would be consistent with the clinical state and not GA (eg increased T cell activation and intermediate monocytes in unstable infants, particularly those born to mothers with chorioamnionitis). Nevertheless, when plotting the change in immune parameters parameters with cGA the graphs were more or less identical to those with postnatal age suggesting time postnatally is the biggest driver of immune trajectory.

- Would also strongly suggest to add range not just interquartiles, as this will provide a more realistic view of the data spread

I have amended all data in the table where a median is report to include the range rather than an interquartile range.

Revised Table 1

	Total Babies (n=39)	Stable Babies (n=10)	Unstable Babies (n=16)	Chorioamnionitis (BCM, n=13)
Gestation, median	28.7 (23.6-31.7)	30.5 (29.1-31.7)	28 (23.6-31.7)	25.7 (24-30.1)
Birthweight, mean	1060g (383)	1366g (242)	1062g (422)	821g (241)
Sex				
Male	23 (59%)	6 (60%)	11 (69%)	6 (46%)
Female	16 (41%)	4 (40%)	5 (31%)	7 (54%)
Multiple births				
Singleton	29 (73%)	5 (50%)	11 (69%)	13 (100%)
Multiple	10 (27%)	5 (50%)	5 (31%)	
Apgar 5 mins, median	9 (2-10)	10 (6-10)	9 (2-10)	9 (2-10)
Maternal ethnicity				
White	9 (23%)	1 (10%)	4 (25%)	4 (31%)
Black	20 (51%)	4 (40%)	10 (63%)	6 (46%)
Asian	7 (18%)	4 (40%)	1 (6%)	2 (15%)
Other	3 (8%)	1 (10%)	1 (6%)	1 (8%)
Antenatal steroids				
Any	36 (92%)	10 (100%)	13 (81%)	13 (100%)
>24 hours before delivery	18 (46%)	6 (60%)	7 (44%)	5 (38%)
None	3 (8%)	0 (0%)	3 (19%)	0 (0%)
Delivery by caesarean				
Yes	24 (62%)	10 (100%)	12 (75%)	2 (15%)
No	15 (38%)	0 (0%)	4 (25%)	11 (85%)
Chorioamnionitis	13 (33%)	0 (0%)	0 (0%)	13 (100%)
Histologically confirmed	12 (31%)			12 (92%)
Antenatal antibiotic administration	11 (28%)	4 (40%)	2 (12%)	5 (38%)
Microbiologically confirmed sepsis (any episode)	14/39 (36%)	0 (0%)	7 (44%)	7 (54%)
Days antibiotics, median, range	15 (3-51)	4 (3-8)	19 (5-51)	19 (5-35)
Suspected GI pathology (including NEC)*	7/39 (18%)	0 (0%)	5 (31%)	2 (15%)
NEC Bell's Stage II or >	3/39 (7%)	0 (0%)	3 (19%)	0 (0%)
Days to full enteral feeds**	14 (7-42)	12 (7-37)	16.5 (8-33)	15 (7-42)
Received any maternal breast milk	37/39 (95%)	10 (100%)	14 (87.5%)	13 (100%)
Abnormalities on CrUSS				
IVH***	6 (15%)	1 (10%)	2 (12.5%)	3 (23%)
HPI	2 (5%)	0	1 (6%)	1 (8%)
PVL	3 (8%)	0	1 (6%)	2 (15%)
Chronic lung disease****	14 (36%)	0 (0%)	8 (50%)	6 (46%)
ROP*****	16 (10%)	1 (10%)	8 (50%)	7 (54%)

Table 2: Data are presented as means (standard deviation); medians (ranges); n=number and percentages.
*Suspected GI pathology is defined as 'any abdominal concerns necessitating nil by mouth for more than 5 days'.
**Full enteral feeds defined as 'the first day on which 100% of fluid volume was administered enterally';
CRUSS=cranial ultrasound; ***IVH= intraventricular haemorrhage defined as 'bleeding contained within and not extending beyond the ventricular system'; HPI= haemorrhagic parenchymal infarction; PVL=periventricular leukomalacia; ****CLD= chronic lung disease defined as 'the need for supplemental oxygen at 36 weeks postmenstrual age'; *****ROP=retinopathy of prematurity defined as 'any stage ROP recorded on formal eye examination'.

- Microbiome analysis: should specify this is ONLY stool analysis, not THE microbiome, we do not know yet what happens correspondingly at other body sites.

This is now made clear in the text.

- I am not sure how the propidium iodide addition would prevent actual bacterial contamination. In principle, PI binds to exposed DNA. It should bind any free DNA and it would enter non-viable cells and bind their DNA. Subsequently, only DNA from initially viable cells is supposed to be amplifiable by PCR due to the PI intercalation. However, as the remaining free PI is washed out, this would not prevent contamination from a) viable cells, b) nor would it prevent contamination from DNA in PCR reagents which would be added after the PI treatment.

To clarify we used propidium monoazide, not propidium iodide. We agree that PMA will not prevent contamination in the PCR process or beyond, but its use in preventing inaccurate results by excluding contamination up to this point, and from non-viable bacteria is still valuable. Contamination in the PCR process is minimised by standard laboratory techniques, and any minor downstream contamination post-PCR will be minimal compared to the millions of amplicons produced.

- They describe differences in diversity and Enterobacteriaceae progression between the stable/unstable groups. Considering demographic differences between these groups, has this been adjusted for GA? Type of milk feeds? Was there probiotic supplementation? What about mode of delivery? Etc.

We were unable to make an adjustment for gestational age between the groups, as the restricted, less extremely premature, age range within our stable group meant that we could not assess the independent impact of gestational age as a confounding factor, nor extrapolate that to the more extremely premature gestations seen within the unstable groups. We could not adjust for type of milk feeds as these were constantly changing in proportion and volume; and the entire stable group was born operatively. No probiotics were used.

Minor

points:

- Use of PMA/ionomycin as sole immune stimulus is concerning. While it often is considered

one of the 'standard' stimuli, it is rather strong and – importantly - entirely unphysiological and may, in my opinion, miss more subtle yet important differences. Why were more specific stimuli (TLR ligands or TCR stimuli) not used?

Whilst we agree that P/I is not physiological, what it does show is which cytokines the cell is capable of making. It also activates a wide range of cell types. TLR /TCR etc stimulation would have been preferable but would have increased the cell numbers required to do separate stimulations and this was a very limiting factor. Furthermore, it is known that signalling downstream of the TCR may be weak in neonates and hence may not have given full activation in order to identify cytokine potential. This was bypassed by the use of P/I.

- The various statements reg. immune functions 'at birth' are factually incorrect. Consent was taken postnatally within 72h, not antenatally. Also, I don't think they looked at cord blood.

We apologise, this has been removed for clarity throughout. Consent was taken *both* antenally and postnatally up to 72 hrs so this statement is correct as is.

- Line 394. This is not unexpected once taking GA at birth into account.
- Faecal sampling 2.3 states daily; 2.7 states every 3 days ? What is it?

Samples were taken (where possible) every day – those selected for analyses were approximately every three days in frequency.

- With this frequency of sampling a huge number of samples should be available. Could they indicate how many in total per infant ? Were they all analysed? If not, why not?

For these 39 babies, we have a total of 1411 samples (median 35/subject); of these only 34 babies underwent microbiome analyses with a total of 1282 samples (median 36/baby). Of these, 713 samples were selected for analysis (median 20 /baby) due to constraints including quality criteria (i.e. samples which we could not isolate sufficient bacterial DNA from were not advanced to sequencing). Samples were collected on a daily basis where available. We selected identified samples for analysis at approximately three day intervals – preliminary work had shown that similar longitudinal gradients would be derived with this reduced sample frequency. The level of sample frequency remains significantly more detailed than comparative studies

- How can it be they only have microbiome analysis in 2/3 of the unstable infants? Does this still support the claim that 'microbiome' is unrelated to immune development ?

Microbiome and immune correlation was performed on individual babies, not as a group, and then the data was grouped together for presentation, hence we feel we can indeed state that there was no correlation in the babies analysed. We were unable to analyse the microbiome on all of the babies on whom the immune system was analysed and hence

could only present those that had both taken. We did have another 3 subjects on which the microbiome had not been initially analysed and have now increased the matched sampling to incorporate these. The graphs and text have been updated accordingly. Having increased the microbiome:immune correlations by these 3 infants, we did not observe an enhanced level of correlation between the two sets of data. One very weak negative correlation did emerge (at the expense of a weak correlation that disappeared!). This further highlights how tenuous these relationships are and our observation that the microbiome was not significantly contributing to the immune development. The new weak negative correlation that appeared with the larger data set-mDCs with *Staphylococcaceae*- is likely to be heavily related to age as mDC increase with age and *Staphylococcaceae* decreases. This is highlighted in the revised text.

- Fig 3A - very hard to make out the important lines, consider change in colour scheme ?

New, clearer copies of the graphs are now in the revised manuscript (now Figure 4).

- Line 343 'most severe infections' - not defined

To avoid confusion this has been removed from the text and expanded in the appropriate figure legend and the text '*Blood culture positive bacteraemias are separated into those from coagulase negative staphylococci and other bacteria as CoNS infections are generally considered to be less severe with lower mortality rates.*

- Demographics - given the relevant differences between groups I suspect the rates of antenatal antibiotic therapy were very different ? This simply must be stated.

These data are now included in the revised Table 1.

Reviewers' comments:

Reviewer #1 (Immune system development, paediatrics) (Remarks to the Author):

Overall the authors have addressed my questions adequately.

Reviewer #2 (Newborn immunology, infection) (Remarks to the Author):

We thank the authors for their effort to improve the previously submitted manuscript. We added our response to their response in the attached.

Response to reviewers

Please find the responses to all the comments made by the reviewers below.

Reviewer #1

Kamdar et al analyze immune cell frequencies and composition in preterm children in relation to clinical features and the microbiome. The topic is of importance, the data seems very interesting, but the data analyses performed are rudimentary and the study feels like a missed opportunity. Several of the main conclusions could have alternative explanations not explored by the authors. Also, neither the data, nor the analysis code seem to be deposited in any public domains, preventing others from reanalyzing the data and me from properly reviewing the work.

We also believe the data are very interesting and we apologise that the methods of data analysis were not clear and hence appeared rudimentary. This was not the case and the exact analysis has been detailed below and in the revised paper. The data will be deposited on the open science framework (OSF) <https://osf.io> and the FCS files will be deposited on the flow repository (<https://flowrepository.org>) post acceptance and prior to publication.

Specific points:

1. In Fig 1, newborn immune features are compared to adults, but adults show a wide range of variation, as expected from previous work (Patin et al, Nat Imm 2018, Roederer et al, Cell 2015, Brodin et al, Cell 2015 etc). What is the relevance of comparing to adults?

Whilst we agree that adults are very variable and we would expect variation between adults and infants, the comparison to adults was merely to highlight which parameters are the most distinct in preterm around the time of birth-hence only a supplementary figure. As our infants were followed longitudinally, we were then able to see which parameters moved towards the levels in adults in the 3 month period which we do feel is a very valid comparison (Fig 2a). This was particularly interesting as some parameters did reach near adult levels whilst others did not which we comment on.

REVIEWER 2: See more on this point below. The missing comparison to term newborns is a key limitation of this manuscript.

Parents would be a more reasonable comparison here given the genetic and environmental relatedness to the newborns in the study. If parents are not available, comparing distributions in newborns to those of adults using Kolmogorov-Smirnoff tests or similar would be more appropriate than the current approach.

Unfortunately, parents were not available. With regards the statistics, we thank the reviewer and acknowledge that a straightforward t test is probably not the most appropriate test in this instance. We have reanalysed the data using a Wilcoxon rank sum

test with a Benjamini-Hochberg correction for multiple comparisons which is more robust given the potential for the data to be non-normally distributed. Whilst the magnitude of the p values has decreased as a result, the overall interpretation of the data has remained unchanged. The revised Supplementary figure 1 is added.

2. Cell frequencies seem to be treated as independent variables throughout the study, when they are in fact highly dependent on each other. As an example, is the B-cell increase due to an actual increase in B-cells or a decrease in some other subsets?

We are aware that many parameters are linked, and where possible we did show actual increases in cell numbers to avoid this (B cells are a good example as questioned by the reviewer, an increase in actual B cell numbers was shown in Supplementary figure 2A). Actual cell numbers are also shown for NKG2D expressing NK cells (Suppl Fig 2C).

REVIEWER 2: Entirely agree with REVIEWER 1 on this. All data presented has to be given as absolute as well as relative proportion. If the authors can not do that, the readers need to know why (i.e. were counting beads not included in all samples to allow absolute quantification? If so, why not, and why was there variability in the ability of assessing absolute numbers between samples (or, as mentioned here by the author) or populations.

In Figure 2 correlations are calculated but these are difficult to interpret given the inherent relationships between features

As we are aware of the reviewer's point regarding variables depending on each other, in this figure we only plotted one parameter (of a correlated pair) if it was inherently related to another parameter (for example, $\gamma\delta$ T cells lacking CD161 expressing IFN- γ were retained whereas the linked IFN- γ negative, CD161 negative $\gamma\delta$ T cell population was removed from the analysis); this was done for all linked populations. Apologies this was unclear, this has now been altered in the text for clarity.

3. The microbiome analysis is also deficient in that global compositional features are not considered and instead only one single family of bacteria is analyzed. Previous studies have shown a patterned progression in preterm gut microbiomes (La Rosa et al, PNAS 2014). How do the authors reconcile the lack of such a patterned progression seen here?

We are aware of the La Rosa paper and its contribution to understanding longitudinal microbiome trends. Perhaps the most crucial differences between their microbiome analyses and ours are that their groups are composed of a heterogeneous mixture of stable and unstable babies, with variable (and from assessment of their study demographics) extensive use of antibiotics. Consequently, the trends they have highlighted are a composite of those of stable and unstable babies. In our cohort, these have been separated to account for antibiotic exposure and therefore minimise the effects of such a potentially powerful modulator of the gut microbiome, something deficient within the wide body of literature pertaining to the preterm gut. They also describe their findings at a class taxonomic level, whereas we have described it at a more detailed family level which makes

comparison difficult. The reason that only Enterobacteriaceae progression is described is that examination of individual subjects' longitudinal trends demonstrated this to be the only taxon which was consistently described across all subjects in our study. The remainder of taxa showed a multimodal distribution of colonisation, meaning that the use of summary statistics (e.g. mean/median) in describing them would have been misleading." We do present the data for select taxa (Enterococcaceae, Bifidobacteriaceae, Staphylococcaceae and Veillonellaceae), but summary measures and statistical analyses on these data would have been inappropriate.

REVIEWER 2: We agree with the concern of REVIEWER 1. While the authors response is relevant (changes in specific taxa or even family are important to highlight rather than only summary stats) it still is necessary to present total composition in order to understand the context of the more specific changes.

4. Some of the microbiome findings seen are consistent with previous reports and could be explained by mode of delivery (High Staph early in life in C-section delivered children). This should be discussed in the manuscript.

The observed association of high Staphylococcaceae abundance in early life in the stable group (uniformly born by operative delivery) is consistent with the previously noted association of staphylococci colonisation after caesarean section (as opposed to natural birth) (Itani et al, 2017, Anaerobe; Shao et al, 2019, Nature). These points are now raised in the manuscript and additional references added.

5. The lack of correlations between microbes and immune features could be due to the failure to consider the pairing of samples from specific individuals. It is my understanding that all immune samples are considered separately, and all microbiome samples considered separately, even though these are in fact linked series of samples from the same individuals. Taking such relationships into account is mandatory in my opinion.

We apologise our analysis was unclear and hence we have elaborated in the manuscript text. For an individual baby, immune measures were mapped to contemporaneous microbiome analysis and a correlation established. The correlation values were then aggregated across all individuals to form the figure. The immune and microbiome samples were NOT considered separately but as linked data within individual babies. The relationships were then pooled to form the figures.

All in all, the authors clearly have a very interesting dataset generated but have failed to capitalize on it and most of the current conclusions are uncertain and not robustly supported by the data. I suggest a complete rework of the analyses, taking feature and sample relationships into account is required to draw meaningful conclusions.

Thank you for the detailed review of our manuscript and data and for the suggestions you have made. We hope we have addressed these in the above responses and highlighted that

we did analyse our data appropriately taking feature and sample relationships into account and have now made this much clearer with apologies that we did not do this sufficiently well initially.

Reviewer #2 (Newborn immunology, microbiome)(Remarks to the Author):

There are several concerns:
Main concerns:

- Lack of term infant comparison - clearly during the neonatal period, this would be the 'gold standard' comparison, not adults as done in the manuscript. Further, with adults there likely will have only been one measurement, ie no longitudinal assessment. Even in term infants massive changes occur in the first few weeks. Although they maybe more dramatic in preterm (as this manuscript posits), comparing them to term newborns would be the appropriate comparator. If this has not been done, please explain why.

As we mention above, comparison to adult was merely to highlight any progression in immune parameters over time towards the levels observed in adults. Within the UK (and likely further afield), it would be difficult to convince an ethics review board to allow repeated sampling in healthy term babies, which was why this was not done as the 'gold' standard comparator group. Preterm infants in hospital are subject to frequent blood tests to monitor their clinical progress. All blood samples taken for this study were taken at the time of routine blood taking for clinical purposes. Indeed, I have contacted a former Chair of a clinical research ethics committee and this was confirmed. While research on healthy (adult) volunteers may be justified by being 'in the public interest', this could not be invoked for healthy babies who very rarely suffer bacterial sepsis. Furthermore, if such a study was approved by a Research Ethics Committee, and indeed the MRC guidelines suggest it is possible, it would be unlikely that parents would agree to routine blood taking for a purpose unrelated to the health of their baby. In addition, the postnatal course for a healthy term baby is not the same as that for a baby born at <32 weeks i.e. it would not be expected to be interrupted by repeated evaluations for suspected infection. Any comparison would therefore be logistically and ethically challenging and of possibly limited value. However, in an attempt to investigate whether the changes observed in our cohort are also mirrored in a term baby cohort, we have analysed different term babies sampled at different ages in the first few months of life. These samples were blood samples from infants prior to cardiac surgery in a separate study on thymus development. As these babies were not followed longitudinally there was, as expected, significant heterogeneity in the different cell subsets, to a similar extent to that seen in the preterm cohort when looking between different babies which all had their own individual profiles (Fig 3C). However, when we looked at several parameters which changed in our preterm cohort over time, interestingly, term babies seem to gain functionality at a similar rate suggesting similar postnatal adaptation in both cohorts. Exceptions to this are discussed. This data has been added as a supplementary data figure (Supplementary Figure 4).

REVIEWER 2: We recognize the difficulty in obtaining healthy, term newborn control groups (although several current cohorts are recruiting term newborns for serial sampling, i.e. while

maybe difficult this is by far not impossible), and thus can accept their reasoning for lack of term newborn controls. However, difficulty in obtaining the appropriate experimental controls does not negate the problems such lack of appropriate controls brings to the interpretation of the data. In short, while we accept their reasoning, the problem persists: Are the dramatic changes observed the 'norm' or peculiar to the pre-term cohort. We are sure the authors would agree that this has massive implications; this has to be clearly stated as a significant limitation of their findings.

- All results in graphs are given in age postnatal weeks, not days. So I cannot quite follow how they can state 'at birth' ? Timing of sampling is hugely important.

We apologise about our use of 'at birth'. The reviewer is correct as this is inaccurate and we have changed this throughout the manuscript, 'at birth' has been removed throughout to avoid any inaccuracies. We do not believe, however that changing the labelling on the graphs into days would have any additional benefit, other than making the axis labels harder to read.

- Storage time is given as an average of 12h and up to 30h at room temperature. Not only is such long storage time known to be a problem (introduces leukocyte adhesion to plastic, degranulation, changes in cytokines, activation marker expression etc.), it also is not clear as to how much difference there was between samples reg. storage. I would only expect much variation for samples taken at the time of sepsis, not the routine weekly samples (as they knew they were coming). Such systematic difference in storage time may have skewed the results.

Unfortunately, as our ethics did not allow collection specifically for this study, even weekly samples were taken only when routine bloods were being taken (often during the night shift) and hence timing was not always convenient. The samples were collected as simultaneously as our ethical permission and clinical opportunity permitted. We fully agree that some populations are lost with extended time frame between sample collection and processing. Indeed, we ran some initial tests on the same blood samples processed at different times to identify such populations. Actually, there were very few of the populations of interest that were affected. Intermediate monocytes were such a population and hence we have added into the manuscript how levels of these are probably underrepresented. As we were wary of this, we did careful observations for any populations that may have appeared decreased etc and checked the timing of the sample. We noted both collection and processing time on each sample so this could be monitored. We excluded any samples that were processed more than 30 hrs after collection from the analyses.

REVIEWER 2: While it helps to know the authors have included the time-to-processing as a variable, to firmly grasp this variable (and potential confounder) it would seem insufficient to us to do this only visually, spot-checking the results. Much preferred would be e.g. a simple linear regression analysis of time to processing vs. each of the particular results.

- The classification of their study subjects is utterly incomplete. For example, it is unclear what the chorioamnionitis status of the 'stable' infants were? It should be similar in distribution as in the unstable infants and should be taken into account for these infants too, since if that were a critical driver of immune responses, it should be present there too.

We apologise that this was unclear. We have clarified this throughout the paper. For ease of understanding the groups are classified as below. There are three groups designated post collection and assessment of clinical course. Those born with histologically confirmed or clinical chorioamnionitis were designated as 1 group. Those infants born in the absence of any evidence of chorioamnionitis were then further subdivided into clinically stable (no antibiotic exposure beyond the first week after birth) and clinically unstable (several repeated episodes of antibiotic treatment in association with suspected infection). With the exception of just one baby, all the babies in the chorioamnionitis group were also clinically unstable requiring several episodes of antibiotic treatment in association with suspected infection. We made this separation in order to assess and indeed overcome the potential confounder presented by the presence of chorioamnionitis.

Furthermore, how is this impacted by the vastly different GA between the groups? At 30w (stable infants) the expected incidence of chorio is <10% whereas at 25w (unstable with chorio group) it more likely was 40-50%. Further lacking is characterization of the degree of placental inflammation (clearly of massive importance with Chorio), including specifically involvement of maternal, fetal or both placental layers. Were Cultures taken and positive?

All the stable babies had no evidence of chorioamnionitis as described above and now made clearer in the manuscript– I would not expect the distribution of chorioamnionitis diagnosis to be the same in the post-hoc allocated stable and unstable groups, as a history of chorioamnionitis is an independent risk factor for complications in the preterm course, thus making them more likely to be retrospectively classified as 'unstable'. Indeed, as now made clear, all but one of the infants born in the context of chorioamnionitis, had an unstable clinical course. I agree that the differing corrected gestational age at birth between the stable and unstable groups is a potentially significant confounder which we now make more reference to in the manuscript.

We provide a table of the chorioamnionitis descriptions for babies in the chorioamnionitis group for the reviewer. Placental swabs were not taken. We do not feel that this detail would be of significant interest to the average reader but can, of course, add this into the manuscript if the editor felt it was of additional value

Subject	Chorioamnionitis Description	Fetal Inflammation Description
18	Severe acute chorioamnionitis	Mild funisitis
39	Acute chorioamnionitis	Umbilical vasculitis (mild-moderate FIRS)
40	Chorioamnionitis	Focal chorionic vasculitis (mild FIRS)

43	Chorioamnionitis	Umbilical vasculitis (moderate FIRS)
44	Acute chorioamnionitis	Early fetal phlebitis
49	Acute chorioamnionitis (severe)	Funisitis (moderate FIRS)
51	Not sent	
59	Acute chorioamnionitis	Vasculitis and funisitis (advanced grade 1 FIRS)
60	Acute chorioamnionitis	No FIRS
66	Acute chorioamnionitis	Vasculitis and funisitis (advanced grade 1 FIRS)
70	Acute chorioamnionitis	Vasculitis and funisitis (intermediate grade 1 FIRS)
76	Acute chorioamnionitis	Vasculitis (mild-moderate FIRS at intermediate stage)
101	Acute chorioamnionitis and	Umbilical phlebitis indicating a maternal and fetal response

- Statistical analysis: the description of immune parameter analysis is a bit brief and doesn't clearly state which confounders were assessed and what parameters were adjusted for

In the microbiome analyses, the biometrics are derived from intra-subject trajectories (independent of gestational age), so this would not have made a difference. The description of the immune parameter analysis has been extended.

REVIEWER 2: In addition to a more detailed description of the analysis, it still is not clear which confounders were assessed (e.g. see above point on time-to-processing) and/or adjusted for.

- Postmenstrual age rather than postnatal age may be better to compare the effects of GA at birth (which is lost if only looking at postnatal time periods).

We have also performed all the analyses using post menstrual age and the graphs are more or less identical to those plotted by postnatal age-this suggests that the changes are driven by environmental effects post birth, this is now fully explained in the text

- Patient demographics: a full flow chart would be advisable; ie how many patients approached, declined, excluded, died etc.

Of the subjects eligible for recruitment, 22% were not approached for consent due to concerns regarding infant short term survival or due to other concerns including capacity to provide consent and availability of the research team. Of the parents approached, 62% agreed that their infants could be enrolled. Stool samples were not collected in just 6% of these infants. This information has been added into the paper and also the figure legend of

the protocol (Fig 1). Of the babies represented in this data set (n=39), none of the babies died.

REVIEWER 2: Could a full regular flow chart (~ clinical trial) not be included as supplemental figure?

Surprising to see no infant with NEC, were these excluded (and yet the stable states outcome GI pathology) ?

Of the five babies in the unstable group three were diagnosed with NEC stage II. These data are updated in the amended table.

REVIEWER: This should also be included in the above mentioned flow chart.

And again: what is the chorioamnionitis status of the 'stable' babies ?

This is addressed above. Any infant with chorioamnionitis is in one group. All but one infant in the chorioamnionitis group had an unstable clinical course.

Comparison between stable and unstable groups is mute - authors state themselves these were more mature, bigger and all born via CS, so not really a control group. At least these parameters need to be adjusted for in analysis. They may simply have been more stable because of these factors. Very likely even looking at the mean GA of the groups might reveal confounders (of note: there seems to be no comparison of e.g. 30.5 vs 28.7 vs 25.4w., which we know are entirely different clinically, even without any tests. This may have massive impact, as either the younger babies die more often (hence request for flow chart).

We agree that the differences in clinical stability may be affected by the differing gestational ages in the cohorts. We would not try to suggest otherwise and indeed, did not suggest that the stable infants are a 'control' group. Comparisons between the groups did highlight differences which would be consistent with the clinical state and not GA (eg increased T cell activation and intermediate monocytes in unstable infants, particularly those born to mothers with chorioamnionitis). Nevertheless, when plotting the change in immune parameters parameters with cGA the graphs were more or less identical to those with postnatal age suggesting time postnatally is the biggest driver of immune trajectory.

REVIEWER 2: This is such an important point (indeed central to the main thrust of this manuscript) that this needs to be clearly displayed and discussed. If there really is no impact of GA, or if any GA-driven difference in immune status is averaged out across postnatal age, this needs to be made crystal clear.

- Would also strongly suggest to add range not just interquartiles, as this will provide a more realistic view of the data spread

I have amended all data in the table where a median is report to include the range rather than an interquartile range.

Revised Table 1

	Total Babies (n=39)	Stable Babies (n=10)	Unstable Babies (n=16)	Chorioamnionitis (BCM, n=13)
Gestation, median	28.7 (23.6-31.7)	30.5 (29.1-31.7)	28 (23.6-31.7)	25.7 (24-30.1)
Birthweight, mean	1060g (383)	1366g (242)	1062g (422)	821g (241)
Sex				
Male	23 (59%)	6 (60%)	11 (69%)	6 (46%)
Female	16 (41%)	4 (40%)	5 (31%)	7 (54%)
Multiple births				
Singleton	29 (73%)	5 (50%)	11 (69%)	13 (100%)
Multiple	10 (27%)	5 (50%)	5 (31%)	
Apgar 5 mins, median	9 (2-10)	10 (6-10)	9 (2-10)	9 (2-10)
Maternal ethnicity				
White	9 (23%)	1 (10%)	4 (25%)	4 (31%)
Black	20 (51%)	4 (40%)	10 (63%)	6 (46%)
Asian	7 (18%)	4 (40%)	1 (6%)	2 (15%)
Other	3 (8%)	1 (10%)	1 (6%)	1 (8%)
Antenatal steroids				
Any	36 (92%)	10 (100%)	13 (81%)	13 (100%)
>24 hours before delivery	18 (46%)	6 (60%)	7 (44%)	5 (38%)
None	3 (8%)	0 (0%)	3 (19%)	0 (0%)
Delivery by caesarean				
Yes	24 (62%)	10 (100%)	12 (75%)	2 (15%)
No	15 (38%)	0 (0%)	4 (25%)	11 (85%)
Chorioamnionitis histologically confirmed	13 (33%)	0 (0%)	0 (0%)	13 (100%)
Antenatal antibiotic administration	11 (28%)	4 (40%)	2 (12%)	5 (38%)
Microbiologically confirmed sepsis (any episode)	14/39 (36%)	0 (0%)	7 (44%)	7 (54%)
Days antibiotics, median, range	15 (3-51)	4 (3-8)	19 (5-51)	19 (5-35)
Suspected GI pathology (including NEC)*	7/39 (18%)	0 (0%)	5 (31%)	2 (15%)
NEC Bell's Stage II or >	3/39 (7%)	0 (0%)	3 (19%)	0 (0%)
Days to full enteral feeds**	14 (7-42)	12 (7-37)	16.5 (8-33)	15 (7-42)
Received any maternal breast milk	37/39 (95%)	10 (100%)	14 (87.5%)	13 (100%)
Abnormalities on CrUSS				
IVH***	6 (15%)	1 (10%)	2 (12.5%)	3 (23%)
HPI	2 (5%)	0	1 (6%)	1 (8%)
PVL	3 (8%)	0	1 (6%)	2 (15%)
Chronic lung disease****	14 (36%)	0 (0%)	8 (50%)	6 (46%)
ROP*****	16 (10%)	1 (10%)	8 (50%)	7 (54%)

Table 2: Data are presented as means (standard deviation); medians (ranges); n=number and percentages.

*Suspected GI pathology is defined as 'any abdominal concerns necessitating nil by mouth for more than 5 days'.

**Full enteral feeds defined as 'the first day on which 100% of fluid volume was administered enterally';

CRUSS=cranial ultrasound; ***IVH= intraventricular haemorrhage defined as 'bleeding contained within and not extending beyond the ventricular system'; HPI= haemorrhagic parenchymal infarction; PVL=periventricular leukomalacia; ****CLD= chronic lung disease defined as 'the need for supplemental oxygen at 36 weeks postmenstrual age'; *****ROP=retinopathy of prematurity defined as 'any stage ROP recorded on formal eye examination'.

- Microbiome analysis: should specify this is ONLY stool analysis, not THE microbiome, we do not know yet what happens correspondingly at other body sites.

This is now made clear in the text.

- I am not sure how the propidium iodide addition would prevent actual bacterial contamination. In principle, PI binds to exposed DNA. It should bind any free DNA and it would enter non-viable cells and bind their DNA. Subsequently, only DNA from initially viable cells is supposed to be amplifiable by PCR due to the PI intercalation. However, as the remaining free PI is washed out, this would not prevent contamination from a) viable cells, b) nor would it prevent contamination from DNA in PCR reagents which would be added after the PI treatment.

To clarify we used propidium monoazide, not propidium iodide. We agree that PMA will not prevent contamination in the PCR process or beyond, but its use in preventing inaccurate results by excluding contamination up to this point, and from non-viable bacteria is still valuable. Contamination in the PCR process is minimised by standard laboratory techniques, and any minor downstream contamination post-PCR will be minimal compared to the millions of amplicons produced.

- They describe differences in diversity and Enterobacteriaceae progression between the stable/unstable groups. Considering demographic differences between these groups, has this been adjusted for GA? Type of milk feeds? Was there probiotic supplementation? What about mode of delivery? Etc.

We were unable to make an adjustment for gestational age between the groups, as the restricted, less extremely premature, age range within our stable group meant that we could not assess the independent impact of gestational age as a confounding factor, nor extrapolate that to the more extremely premature gestations seen within the unstable groups. We could not adjust for type of milk feeds as these were constantly changing in proportion and volume; and the entire stable group was born operatively. No probiotics were used.

REVIEWER 2: These limitations need to be clearly stated in the manuscript.

Minor points:

- Use of PMA/ionomycin as sole immune stimulus is concerning. While it often is considered one of the 'standard' stimuli, it is rather strong and – importantly - entirely unphysiological and may, in my opinion, miss more subtle yet important differences. Why were more specific stimuli (TLR ligands or TCR stimuli) not used?

Whilst we agree that P/I is not physiological, what it does show is which cytokines the cell is capable of making. It also activates a wide range of cell types. TLR /TCR etc stimulation would have been preferable but would have increased the cell numbers required to do

separate stimulations and this was a very limiting factor. Furthermore, it is known that signalling downstream of the TCR may be weak in neonates and hence may not have given full activation in order to identify cytokine potential. This was bypassed by the use of P/I.

- The various statements reg. immune functions 'at birth' are factually incorrect. Consent was taken postnatally within 72h, not antenatally. Also, I don't think they looked at cord blood.

We apologise, this has been removed for clarity throughout. Consent was taken *both* antenally and postnatally up to 72 hrs so this statement is correct as is.

- Line 394. This is not unexpected once taking GA at birth into account.
- Faecal sampling 2.3 states daily; 2.7 states every 3 days ? What is it?

Samples were taken (where possible) every day – those selected for analyses were approximately every three days in frequency.

- With this frequency of sampling a huge number of samples should be available. Could they indicate how many in total per infant ? Were they all analysed? If not, why not?

For these 39 babies, we have a total of 1411 samples (median 35/subject); of these only 34 babies underwent microbiome analyses with a total of 1282 samples (median 36/baby). Of these, 713 samples were selected for analysis (median 20 /baby) due to constraints including quality criteria (i.e. samples which we could not isolate sufficient bacterial DNA from were not advanced to sequencing). Samples were collected on a daily basis where available. We selected identified samples for analysis at approximately three day intervals – preliminary work had shown that similar longitudinal gradients would be derived with this reduced sample frequency. The level of sample frequency remains significantly more detailed than comparative studies

- How can it be they only have microbiome analysis in 2/3 of the unstable infants? Does this still support the claim that 'microbiome' is unrelated to immune development ?

Microbiome and immune correlation was performed on individual babies, not as a group, and then the data was grouped together for presentation, hence we feel we can indeed state that there was no correlation in the babies analysed. We were unable to analyse the microbiome on all of the babies on whom the immune system was analysed and hence could only present those that had both taken. We did have another 3 subjects on which the microbiome had not been initially analysed and have now increased the matched sampling to incorporate these. The graphs and text have been updated accordingly. Having increased the microbiome:immune correlations by these 3 infants, we did not observe an enhanced level of correlation between the two sets of data. One very weak negative correlation did emerge (at the expense of a weak correlation that disappeared!). This further highlights how tenuous these relationships are and our observation that the microbiome was not

significantly contributing to the immune development. The new weak negative correlation that appeared with the larger data set-mDCs with *Staphylococcaceae*- is likely to be heavily related to age as mDC increase with age and *Staphylococcaceae* decreases. This is highlighted in the revised text.

- Fig 3A - very hard to make out the important lines, consider change in colour scheme ?

New, clearer copies of the graphs are now in the revised manuscript (now Figure 4).

- Line 343 'most severe infections' - not defined

To avoid confusion this has been removed from the text and expanded in the appropriate figure legend and the text '*Blood culture positive bacteraemias are separated into those from coagulase negative staphylococci and other bacteria as CoNS infections are generally considered to be less severe with lower mortality rates.*

- Demographics - given the relevant differences between groups I suspect the rates of antenatal antibiotic therapy were very different ? This simply must be stated.

These data are now included in the revised Table 1.

Response to reviewers Part 2

Please find the responses to all the additional comments made by reviewer 2 below in green. We note that reviewer 1 stated that 'Overall the authors have addressed my questions adequately'. We therefore, comment only to the additional comments from reviewer 2 who had also agreed we had improved our previously submitted manuscript. For ease of understanding, we have removed any comments on which no further action is required.

We are aware of the journal policy regarding reviewer comments and the publication of these comments and the responses. As such, we feel that the addition of much of the new data below into the manuscript would itself not change the message significantly (indeed there is not space to do this) and feel that the publication of its content as part of the supplementary data sets would suffice.

Comments made by Reviewer 1 on which reviewer 2 now comments:

1. In Fig 1, newborn immune features are compared to adults, but adults show a wide range of variation, as expected from previous work (Patin et al, Nat Imm 2018, Roederer et al, Cell 2015, Brodin et al, Cell 2015 etc). What is the relevance of comparing to adults?

Whilst we agree that adults are very variable and we would expect variation between adults and infants, the comparison to adults was merely to highlight which parameters are the most distinct in preterm around the time of birth-hence only a supplementary figure. As our infants were followed longitudinally, we were then able to see which parameters moved towards the levels in adults in the 3 month period which we do feel is a very valid comparison (Fig 2a). This was particularly interesting as some parameters did reach near adult levels whilst others did not which we comment on.

REVIEWER 2: See more on this point below. The missing comparison to term newborns is a key limitation of this manuscript.

We address this point below

2. Cell frequencies seem to be treated as independent variables throughout the study, when they are in fact highly dependent on each other. As an example, is the B-cell increase due to an actual increase in B-cells or a decrease in some other subsets?

We are aware that many parameters are linked, and where possible we did show actual increases in cell numbers to avoid this (B cells are a good example as questioned by the reviewer, an increase in actual B cell numbers was shown in Supplementary figure 2A). Actual cell numbers are also shown for NKG2D expressing NK cells (Suppl Fig 2C).

REVIEWER 2: Entirely agree with REVIEWER 1 on this. All data presented has to be given as absolute as well as relative proportion. If the authors can not do that, the readers need to know why (i.e. were counting beads not included in all samples to allow absolute quantification? If so, why not, and why was there variability in the ability of assessing absolute numbers between samples (or, as mentioned here by the author) or populations.

We note that our focus on percentage representation does actually reflect common practice in immune monitoring studies (Sobolev, O. *et al. Nat. Immunol.* 17, 204–213 (2016); Tsang, J. S. *et al. Cell* 157, 499–513 (2014); Brodin, P. *et al. Cell.* 2015 Jan 15;160(1-2):37-47). We also note that in a study by one of our co-authors (in press, *Nat Immunol* 2019, bioRxiv 688010; doi: <https://doi.org/10.1101/688010>), out of peripheral blood data from c.12,000 mice, the values for cell counts are extremely variable relative to cell proportion as shown in the figure below, suggesting the latter to be more representative.

Nevertheless, we have added further examples of actual cell numbers of distinct populations into Suppl Fig 2. We now show absolute numbers of B cells, NK cells, $\gamma\delta$ T cells and CD8 T cells (Supplementary Figure 2). Obviously, absolute number variations in cell surface markers/cytokine production on cells will be dependent on the basic cell numbers and hence these are the number plots shown.

3. The microbiome analysis is also deficient in that global compositional features are not considered and instead only one single family of bacteria is analyzed. Previous studies have shown a patterned progression in preterm gut microbiomes (La Rosa et al, PNAS 2014). How do the authors reconcile the lack of such a patterned progression seen here?

We are aware of the La Rosa paper and its contribution to understanding longitudinal microbiome trends. Perhaps the most crucial differences between their microbiome analyses and ours are that their groups are composed of a heterogeneous mixture of stable and unstable babies, with variable (and from assessment of their study demographics) extensive use of antibiotics. Consequently, the trends they have highlighted are a composite of those of stable and unstable babies. In our cohort, these have been separated to account for antibiotic exposure and therefore minimise the effects of such a potentially

powerful modulator of the gut microbiome, something deficient within the wide body of literature pertaining to the preterm gut. They also describe their findings at a class taxonomic level, whereas we have described it at a more detailed family level which makes comparison difficult. The reason that only Enterobacteriaceae progression is described is that examination of individual subjects' longitudinal trends demonstrated this to be the only taxon which was consistently described across all subjects in our study. The remainder of taxa showed a multimodal distribution of colonisation, meaning that the use of summary statistics (e.g. mean/median) in describing them would have been misleading." We do present the data for select taxa (Enterococcaceae, Bifidobacteriaceae, Staphylococcaceae and Veillonellaceae), but summary measures and statistical analyses on these data would have been inappropriate.

REVIEWER 2: We agree with the concern of REVIEWER 1. While the authors response is relevant (changes in specific taxa or even family are important to highlight rather than only summary stats) it still is necessary to present total composition in order to understand the context of the more specific changes.

In view of the changing nature of the preterm intestinal microbiome, we felt that summary measures of composition would not be representative, and consequently, we presented dynamic measures of microbiome development. The range of taxa over which summary dynamic measures were appropriate was limited to those taxa which were widely distributed across samples. However, in view the reviewer's request for data on total composition, we present the weighted (by sample number) mean abundance of all family-level taxa present in the three clinical cohorts; these cohort-level summary statistics are derived from the mean abundance across all samples per individual subject. Data are presented as both a stacked bar chart (A) and table(B):

(A) Stacked Bar Chart Total Composition

(B) Table

Family-level taxon	Mean abundance across all subjects (% , range)	Mean abundance in stable group (% , range)	Mean abundance in chorioamnionitis group (% , range)	Mean abundance in unstable, non-chorioamnionitis group (% , range)
Enterobacteriaceae	68.8 (1.2-93.5)	65.4 (47.1-88.7)	68.6 (1.2-93.5)	71.5 (8.4-88.3)
Staphylococcaceae	9.5 (0.2-68.6)	8.3 (0.2-24.4)	8.6 (1.2-47.8)	11.5 (0.2-68.6)
Enterococcaceae	7.1 (0-34.2)	3.2 (0.4-9.4)	10.9 (0.1-34.2)	5.6 (0-28.9)
Veillonellaceae	5.8 (0.1-20.1)	9.5 (0.1-20.1)	3.6 (0.1-10.6)	5.9 (0.1-13.3)
Bifidobacteriaceae	2 (0-12.3)	4.4 (0-9.9)	1.6 (0-12.3)	0.7 (0-3.6)
Clostridiaceae	1.6 (0-9.7)	2 (0-5)	2.1 (0-9.7)	0.8 (0-3.6)
Streptococcaceae	1.3 (0-33.4)	1.7 (0-14.7)	1.6 (0-33.4)	0.6 (0-3.1)
Pasteurellaceae	0.5 (0-4.2)	0.4 (0-3.7)	0.4 (0-4.2)	0.6 (0-1.9)
Bacteroidaceae	0.4 (0-19.6)	1.7 (0-19.6)	0 (0-0.1)	0 (0-0.1)
Lactobacillaceae	0.4 (0-3.9)	0.9 (0-3.9)	0 (0-0.4)	0.4 (0-3.5)
Peptoniphilaceae	0.4 (0-1.4)	0.6 (0-1.4)	0.3 (0-1.3)	0.2 (0-0.9)
Fusobacteriaceae	0.3 (0-7.1)	1.1 (0-7.1)	0 (0-0)	0 (0-0.1)
Actinomycetaceae	0.3 (0-10.8)	0.1 (0-0.4)	0 (0-0)	0.7 (0-10.8)
Pseudomonadaceae	0.3 (0-5.4)	0.2 (0-1.4)	0 (0-0)	0.7 (0-5.4)
Acidaminococcaceae	0.3 (0-3.9)	0 (0-0)	0.4 (0-3.9)	0.2 (0-3.5)
Moraxellaceae	0.3 (0-2.6)	0.1 (0-0.3)	0.5 (0-2.6)	0.1 (0-1.2)
Prevotellaceae	0.2 (0-2.8)	0 (0-0)	0.4 (0-2.8)	0 (0-0.2)
Leptotrichiaceae	0.2 (0-2.8)	0 (0-0)	0.4 (0-2.8)	0 (0-0)
Corynebacteriaceae	0.1 (0-0.9)	0 (0-0.1)	0.2 (0-0.9)	0.1 (0-0.5)
Peptostreptococcaceae	0.1 (0-1.4)	0.1 (0-0.8)	0.1 (0-1.4)	0 (0-0.1)
Micrococcaceae	0.1 (0-0.6)	0 (0-0.1)	0 (0-0.1)	0.1 (0-0.6)
Bacillales_incertae_sedis	0 (0-0.1)	0 (0-0)	0.1 (0-0.1)	0 (0-0.1)
Lachnospiraceae	0 (0-1.1)	0 (0-0)	0.1 (0-1.1)	0 (0-0.1)
Dermabacteraceae	0 (0-0.6)	0 (0-0.2)	0.1 (0-0.6)	0 (0-0)
Coriobacteriaceae	0 (0-0.4)	0.1 (0-0.4)	0 (0-0.2)	0 (0-0)
Aerococcaceae	0 (0-0.4)	0 (0-0)	0 (0-0)	0 (0-0.4)
Campylobacteraceae	0 (0-0.2)	0 (0-0)	0 (0-0.2)	0 (0-0)
Xanthomonadaceae	0 (0-0.1)	0 (0-0)	0 (0-0)	0 (0-0.1)
Neisseriaceae	0 (0-0)	0 (0-0)	0 (0-0)	0 (0-0)
Porphyromonadaceae	0 (0-0)	0 (0-0)	0 (0-0)	0 (0-0)
Burkholderiaceae	0 (0-0)	0 (0-0)	0 (0-0)	0 (0-0)

Comments by reviewer 2:

There are several concerns:

Main concerns:

- Lack of term infant comparison - clearly during the neonatal period, this would be the 'gold standard' comparison, not adults as done in the manuscript. Further, with adults there likely will have only been one measurement, ie no longitudinal assessment. Even in term infants massive changes occur in the first few weeks. Although they may be more dramatic in preterm (as this manuscript posits), comparing them to term newborns would be the appropriate comparator. If this has not been done, please explain why.

As we mention above, comparison to adult was merely to highlight any progression in immune parameters over time towards the levels observed in adults. Within the UK (and likely further afield), it would be difficult to convince an ethics review board to allow repeated sampling in healthy term babies, which was why this was not done as the 'gold' standard comparator group. Preterm infants in hospital are subject to frequent blood tests to monitor their clinical progress. All blood samples taken for this study were taken at the time of routine blood taking for clinical purposes. Indeed, I have contacted a former Chair of a clinical research ethics committee and this was confirmed. While research on healthy (adult) volunteers may be justified by being 'in the public interest', this could not be invoked for healthy babies who very rarely suffer bacterial sepsis. Furthermore, if such a study was approved by a Research Ethics Committee, and indeed the MRC guidelines suggest it is possible, it would be unlikely that parents would agree to routine blood taking for a purpose unrelated to the health of their baby. In addition, the postnatal course for a healthy term baby is not the same as that for a baby born at <32 weeks i.e. it would not be expected to be interrupted by repeated evaluations for suspected infection. Any comparison would therefore be logistically and ethically challenging and of possibly limited value. However, in an attempt to investigate whether the changes observed in our cohort are also mirrored in a term baby cohort, we have analysed different term babies sampled at different ages in the first few months of life. These samples were blood samples from infants prior to cardiac surgery in a separate study on thymus development. As these babies were not followed longitudinally there was, as expected, significant heterogeneity in the different cell subsets, to a similar extent to that seen in the preterm cohort when looking between different babies which all had their own individual profiles (Fig 3C). However, when we looked at several parameters which changed in our preterm cohort over time, interestingly, term babies seem to gain functionality at a similar rate suggesting similar postnatal adaptation in both cohorts. Exceptions to this are discussed. This data has been added as a supplementary data figure (Supplementary Figure 4).

REVIEWER 2: We recognize the difficulty in obtaining healthy, term newborn control groups (although several current cohorts are recruiting term newborns for serial sampling, i.e. while maybe difficult this is by far not impossible), and thus can accept their reasoning for lack of term newborn controls. However, difficulty in obtaining the appropriate experimental controls does not negate the problems such lack of appropriate controls brings to the interpretation of the data. In short, while we accept their reasoning, the problem persists: Are the dramatic changes observed the 'norm' or

peculiar to the pre-term cohort. We are sure the authors would agree that this has massive implications; this has to be clearly stated as a significant limitation of their findings.

We are grateful that the reviewer appreciates this is a difficult cohort to obtain. We believe, as did reviewer 1, that the smaller term cohort, whilst heterogeneous, does add important information into the current data set suggesting the changes may be the 'norm'. Nevertheless, a term cohort would be the ideal and hence we now clearly state in the manuscript that the lack of a similarly matched longitudinal study in term infants is a limitation of the study. 'The interpretation of our data is however constrained by the lack of an equivalent longitudinal dataset from a cohort of healthy full term infants to which to compare our findings. In our experience no research ethics committee would sanction such extensive sampling in healthy term babies to the same degree of sampling that we have conducted in patients recruited to our study. To address this we, like others, have analysed samples obtained opportunistically from hospitalised infants, in our case awaiting cardiac surgery; the normality of which can be challenged'.

- Storage time is given as an average of 12h and up to 30h at room temperature. Not only is such long storage time known to be a problem (introduces leukocyte adhesion to plastic, degranulation, changes in cytokines, activation marker expression etc.), it also is not clear as to how much difference there was between samples reg. storage. I would only expect much variation for samples taken at the time of sepsis, not the routine weekly samples (as they knew they were coming). Such systematic difference in storage time may have skewed the results.

Unfortunately, as our ethics did not allow collection specifically for this study, even weekly samples were taken only when routine bloods were being taken (often during the night shift) and hence timing was not always convenient. The samples were collected as simultaneously as our ethical permission and clinical opportunity permitted. We fully agree that some populations are lost with extended time frame between sample collection and processing. Indeed, we ran some initial tests on the same blood samples processed at different times to identify such populations. Actually, there were very few of the populations of interest that were affected. Intermediate monocytes were such a population and hence we have added into the manuscript how levels of these are probably underrepresented. As we were wary of this, we did careful observations for any populations that may have appeared decreased etc and checked the timing of the sample. We noted both collection and processing time on each sample so this could be monitored. We excluded any samples that were processed more than 30 hrs after collection from the analyses.

REVIEWER 2: While it helps to know the authors have included the time-to-processing as a variable, to firmly grasp this variable (and potential confounder) it would seem insufficient to us to do this only visually, spot-checking the results. Much preferred would be e.g. a simple linear regression analysis of time to processing vs. each of the particular results.

We are pleased that the reviewer has noted our scientific rigour by recording these data when many would not have done so. We have performed a regression analysis as suggested

and note that some variables are, as expected, significantly associated with time to processing. For example, the ability to make cytokines upon stimulation is negatively associated with processing time. We include a table below of the basic parameters most associated ($R > +0.2$, $R < -0.2$) with delays in processing. Nevertheless, when we assessed any relationship between processing time and the sample age (considering we are assessing changes in parameters over time) there was none, further highlighting that our data suggests the development of immune parameters is related to post-natal age. We have added a sentence to the paper regarding this 'We observed that the time to blood processing did have an adverse effect on cytokine production. Nevertheless, when we assessed any relationship between processing time and sample age (considering we are assessing changes in parameters over time) there was none, further highlighting our data that suggests the development of immune parameters is related to post-natal age'.

NK IFN γ +	-0.43090375
CD8+ IFN γ +	-0.401543301
NKG2d+ $\gamma\delta$ cells	-0.363380927
NKG2D NK cells	-0.335913996
CD8+ IL-8+	-0.321397612
CD8+ IL-2+	-0.292208787
CD56 bright NK cells	-0.289726445
CD3+ V δ 2+	-0.28367158
CD4+ IL-13+	-0.266213386

CD4 IL-2+	-0.262222132
CD8+ NKG2D+ cells	-0.261472639
CD4+ IL-8+	-0.261410079
NK TNF α +	-0.252093221
Activated Treg	-0.23334615
$\gamma\delta$ + CD161+ cells	-0.219192188
$\gamma\delta$ IFN γ +	-0.215078142
CD4+ IFN γ +	-0.214883675
% NK cells	-0.210838591
NK IL-8+	0.230519874
NK IL-13+	0.246188345
activated CD69+ CD4 T cells	0.279576743
activated CD69+ $\gamma\delta$ + T cells	0.323899774
NK IL-8+	0.230519874

- Statistical analysis: the description of immune parameter analysis is a bit brief and doesn't clearly state which confounders were assessed and what parameters were adjusted for

In the microbiome analyses, the biometrics are derived from intra-subject trajectories (independent of gestational age), so this would not have made a difference. The description of the immune parameter analysis has been extended.

REVIEWER 2: In addition to a more detailed description of the analysis, it still is not clear which confounders were assessed (e.g. see above point on time-to-processing) and/or adjusted for.

We have extended our analyses to include the time to processing which we believe we have comprehensively addressed in the explanation we have provided above. Furthermore, we now provide additional figures of sample numbers as requested. With respect to clinical data that may confound these analyses, you will note from previous submitted versions of our manuscript that we have (from the outset) stated the potential confounder of gestational age in our data set and we would like to draw your attention to the following statements in the manuscript:

1. *Of note, babies in the stable group were generally born at later gestation, had higher birth weights and all were delivered by caesarean section (Section 3.1).*
2. *The observation that CXCL8 and TNF- α production may be reciprocally associated with GA²² and that unstable infants in this cohort were generally born at an earlier GA, may also be an explanation (section 4: Discussion).*
3. *We cannot exclude the potential confounder of GA in our data set as all of the stable babies were born at a later GA (section 4: Discussion).*

- Patient demographics: a full flow chart would be advisable; ie how many patients approached, declined, excluded, died etc.

Of the subjects eligible for recruitment, 22% were not approached for consent due to concerns regarding infant short term survival or due to other concerns including capacity to provide consent and availability of the research team. Of the parents approached, 62%

agreed that their infants could be enrolled. Stool samples were not collected in just 6% of these infants. This information has been added into the paper and also the figure legend of the protocol (Fig 1). Of the babies represented in this data set (n=39), none of the babies died.

REVIWER 2: Could a full regular flow chart (~ clinical trial) not be included as supplemental figure?

We have altered Figure 1 to include the percentages of successfully recruited participants into the study and as such have presented our data in a manner that is similar to previously published work in this area (Olin et al, 2018) and have therefore not produced a trial consort diagram.

REVIEWERS' COMMENTS:

Reviewer #2 (Remarks to the Author):

The authors have responded to all requests, and importantly emphasized the limitations of the ability to interpret some of the data presented.